# Breaking Barriers to an HIV-1 Cure: Innovations in Gene Editing, Immune Modulation, and Reservoir Eradication

**DOI:** 10.3390/life15020276

**Published:** 2025-02-11

**Authors:** Ana Borrajo

**Affiliations:** Department of Microbiology and Parasitology, Faculty of Pharmacy, Complutense University of Madrid, 28040 Madrid, Spain; anborraj@ucm.es

**Keywords:** HIV-1, gene-editing, CAR T-cell therapy, reservoir, CRISPR/Cas9

## Abstract

Recent advances in virology, particularly in the study of HIV-1, have significantly progressed the pursuit of a definitive cure for the disease. Emerging therapeutic strategies encompass innovative gene-editing technologies, immune-modulatory interventions, and next-generation antiretroviral agents. Efforts to eliminate or control viral reservoirs have also gained momentum, with the aim of achieving durable viral remission without the continuous requirement for antiretroviral therapy. Despite these promising developments, critical challenges persist in bridging the gap between laboratory findings and clinical implementation. This review provides a comprehensive analysis of recent breakthroughs, ongoing clinical trials, and the barriers that must be addressed to translate these advancements into effective treatments, emphasizing the multifaceted approaches being pursued to achieve a curative solution for HIV-1 infection.

## 1. Introduction

Over four decades since the discovery of HIV-1 as the causative agent of AIDS, the field of HIV research has made extraordinary progress. The UNAIDS 2024 Fact Sheet provides updated statistics on HIV prevalence in 2023: approximately 39.9 million individuals globally were living with HIV in that year, there were 1.3 million new HIV infections, and 630,000 people died from AIDS-related illnesses. Additionally, 30.7 million people were accessing antiretroviral therapy (ART) [1]. Advances in ART have transformed what was once a fatal disease into a manageable chronic condition, dramatically improving life expectancy and quality of life for millions of individuals worldwide. The efficacy of ART has been demonstrated in suppressing active viral replication and reducing viral loads in plasma to undetectable levels [2]. Despite transforming the clinical management of HIV-1 and effectively suppressing viral replication, ART has not achieved a definitive cure due to the persistence of a latent viral reservoir, which remains a major barrier to eradication [3]. Therefore, HIV-1 continues to pose a major global health challenge, with over 38 million individuals living with the virus and approximately 1.3 million new infections occurring annually.

Lifelong treatment remains necessary, as the virus persists in latent reservoirs—cells where HIV integrates into the host genome—evading immune responses and conventional therapies [2]. Latent reservoirs refer to a population of long-lived cells, primarily resting memory CD4+ T cells, that harbor integrated but transcriptionally silent HIV provirus. These reservoirs persist despite ART because the virus remains in a dormant state, evading immune detection and viral replication. Upon cellular activation, the latent virus can reactivate, leading to viral rebound if ART is interrupted. Latent reservoirs represent a major barrier to HIV-cure-finding efforts.

In recent years, significant breakthroughs in HIV-1 research have shifted the paradigm from merely controlling the virus to pursuing its complete eradication. Innovative approaches, including the “shock-and-kill” and “block-and-lock” strategies, have been designed to target latent reservoirs, while the emergence of gene-editing technologies such as Clustered Regularly Interspaced Short Palindromic Repeats (CRISPR)-Cas9 offers the potential for permanently excising integrated proviruses [4]. Furthermore, immune-based therapies, such as broadly neutralizing antibodies (bNAbs) and therapeutic vaccines, are demonstrating promise not only in achieving viral suppression but also in eradication efforts [5]. Advances in our understanding of HIV-1 virology and immunology, coupled with the application of state-of-the-art methodologies like single-cell transcriptomics and artificial intelligence, are accelerating progress toward the ultimate goal of a definitive cure.

This review examines the current advancements in key complementary and alternative strategies aimed at targeting the viral reservoir. Summaries and comparisons are provided for each major approach. All the strategies reviewed present distinct advantages and promising results, emphasizing their potential for further exploration. Among these, CRISPR/Cas9 stands out by addressing a specific niche, highlighting its potential to make a unique contribution to the therapeutic options available. CRISPR/Cas9 has been used to excise latent HIV Deoxyribonucleic acid (DNA) integrated into the host cell genome, representing a unique strategy for eradicating the virus rather than simply suppressing it with ART.

Ongoing challenges in the scientific progress of the research field include addressing the evolution of viral resistance, immune evasion, and the heterogeneity of patient responses to treatments complicate the development of universally effective therapies. Collaborative research efforts are needed to overcome these hurdles and explore innovative therapeutic approaches, including gene editing technologies like CRISPR/Cas9, immune modulation, and combination therapies. Through the examination of these innovations, it is evident that scientific progress is bringing us closer to achieving a definitive cure, offering hope to millions affected by this global epidemic.

## 2. Relevant Section and Discussion: Emerging Strategies for Targeting the HIV-1 Viral Reservoirs

The persistent viral reservoir of HIV-1 remains a formidable barrier to achieving a functional cure, as current ART can suppress viral replication but fail to eradicate latently infected cells. Consequently, diverse therapeutic approaches have been developed and investigated to specifically target and reduce the viral reservoir, each offering unique advantages and showing promising results. These approaches include gene editing tools like CRISPR/Cas systems [6], chimeric antigen receptor (CAR) T-cell therapy [7], latency-reversing agents (LRAs) [8], bNAbs [9], and immunotherapeutic strategies that aim to eliminate or control HIV-1-infected cells [10].

Each method addresses critical aspects of HIV-1 persistence and reactivation. For example, gene-editing technologies such as CRISPR/Cas9 hold the potential to excise proviral DNA from infected cells or disrupt host genes required for viral entry, such as C-C chemokine receptor type five (CCR5) and C-X-C chemokine receptor type four (CXCR4) [6]. CAR T-cell therapy [7], adapted from cancer immunotherapy, enables the targeted elimination of HIV-1-infected cells through T cells engineered to recognize specific viral antigens. LRAs, on the other hand, are drugs that reactivate viral transcription in the latently infected cells within the reservoir exposing these cells to immune clearance mechanisms or cytotoxic agents [8]. LRAs with demonstrated potency in cells from people living with HIV-1 have been hypothesized to reactivate latency through different mechanisms of action [8]. BNAbs offer another avenue for targeting viral reservoirs, particularly by neutralizing diverse strains of HIV-1 and recruiting effector immune responses [9]. Additionally, immunotherapies, including therapeutic vaccines, have been explored to enhance immune surveillance and the clearance of infected cells [10]. Each approach presents distinct advantages. Gene editing offers precision and permanence, CAR T-cell therapy has demonstrated efficacy in in vivo models, and LRAs enable viral reactivation, which is crucial for targeting latent reservoirs [11]. Nonetheless, challenges remain, including off-target effects, persistence and expansion of engineered cells, toxicity, and the heterogeneity of the viral reservoir. Comparatively evaluating these strategies provides valuable insights into their strengths and limitations and highlights areas requiring further optimization [11].

LRAs can be categorized into several classes based on their mechanisms of action. The first class includes epigenetic modifiers, which reverse the repressive epigenetic marks around the integrated provirus that influence HIV transcription. The histone methyltransferase (HMT) inhibitors and histone deacetylase inhibitors (HDACi) are the more-studied inhibitors; these reverse the repressive acetyl and methyl marks on the integrated HIV genome, its surrounding DNA, and the associated histone tails in nucleosomes [12,13]. BRG1/BRM-associated factor (BAF) inhibitors alter the positioning of nucleosomes on integrated HIV DNA, facilitating transcription of the HIV genome [14]. An interesting study shows that these BAF complexes facilitate the access of transcription factors and cofactors to enhancers and promoters, thereby regulating gene expression, which is critical for the growth and differentiation of acute myeloid leukemia stem and progenitor cells. In acute myeloid leukemia with MLL1 rearrangement (MLL1r) or mutant NPM1 (mtNPM1), treatment with menin inhibitors (MIs) can induce clinical remission; however, many patients either exhibit resistance or experience relapse, with some acquiring menin mutations. FHD-286—an orally bioavailable and selective BRG1/BRM inhibitor currently in clinical development for acute myeloid leukemia—has demonstrated potent effects in preclinical studies. FHD-286 induces differentiation and lethality in AML cells with MLL1r or mtNPM1 by disrupting chromatin accessibility and downregulating key factors such as c-Myc, PU.1, and CDK4/6. Combining FHD-286 with decitabine, a bromodomain and extra-terminal domain (BET) inhibitor, a menin inhibitor or venetoclax significantly enhanced acute myeloid leukemia cell lethality in vitro in a synergistic manner. In patient-derived xenograft models of acute myeloid leukemia with MLL1r or mtNPM1, FHD-286 treatment effectively reduced leukemia burden, prolonged survival, and diminished the leukemia-initiating potential of acute myeloid leukemia stem and progenitor cells. Furthermore, co-treatment of FHD-286 with BET inhibitors, menin inhibitors, decitabine, or venetoclax significantly reduced disease burden and improved survival outcomes compared to single-agent therapies, without inducing notable toxicity. These findings underscore the potential of FHD-286-based combination therapies as a promising treatment strategy for acute myeloid leukemia with MLL1r or mtNPM1 [14].

The second class, intracellular signaling modulators, consists of drugs that regulate protein kinases in signaling pathways, influencing the binding of transcription factors (TFs) to long terminal repeats (LTRs). These include protein kinase C (PKC) agonists and compounds within the PI3K/Akt or JAK/STAT pathways [13]. Additionally, second mitochondria-derived activator of caspases (SMAC) mimetics can be employed to inhibit the degradation of NF-κB-inducing kinase (NIK), thereby promoting the accumulation of NF-κB [15]. Other classes of LRAs include cytokine or immune receptor agonists, which stimulate immune cells through interleukins (ILs), cytokines, T-cell receptors (TCRs), checkpoint inhibitors, or Toll-like receptor (TLR) agonists. Once transcription is initiated, transcription elongation factors can be used to enhance the activity of Tat, which is crucial for transcription elongation [16]. Notable examples in this class include BET inhibitors, which antagonize the inhibitor of P-TEFb, thereby activating the recruitment of P-TEFb to the LTR [12]. Finally, an unclassified group of LRAs includes previously used drugs that have been found to reactivate HIV, although their mechanisms of action remain unclear [13].

This summary and comparison of major approaches will underscore their therapeutic potential, emphasizing the need for integrative strategies and continued exploration to address the complexities of targeting HIV-1 reservoirs (Table 1). A comprehensive understanding of these methods and their synergistic potential will be pivotal in advancing toward a functional cure for HIV-1.

### 2.1. Shock-and-Kill Method

The persistence of latent reservoirs represents the primary barrier to achieving an HIV cure, as mentioned above. The latently infected cells are not detected by the immune system, which impedes their recognition and elimination via immune-mediated clearance or direct viral cytolysis [6]. The reactivation of HIV using LRAs, which target the mechanisms responsible for latency (“shock”), would induce HIV RNA synthesis and viral protein production. This reactivation would then allow these cells to be recognized and eliminated (“kill”) by host immune defense mechanisms or through viral cytolysis [13]. The shock-and-kill strategy aims to utilize LRAs to activate HIV-1 gene expression, enabling host CD8+ T cells to identify and eliminate HIV-1-infected cells within latent reservoirs [17]. Promising preliminary results have been obtained with this approach. HDACi have been shown to reverse latency and induce HIV-1 RNA expression in models such as Simian Immunodeficiency Virus (SIV)-infected rhesus macaques [18], in clinical trials [19], and in ex vivo patient-derived CD4+ T cells [20]. However, a key limitation of the shock-and-kill approach is that HDACi have shown limited or no effect on the size of the latent reservoir, both in ex vivo and in vivo studies, as well as in clinical trials [21]. While in vitro studies and animal models suggest that HDACi, such as Suberoylanilide Hydroxamic Acid (SAHA) and panobinostat, can induce HIV transcription and reduce latency, clinical outcomes have been less promising [21]. For instance, despite achieving sustained serum levels with a long-acting injectable system, SAHA did not significantly impact plasma viral RNA or viral suppression. Instead, it increased cell-associated HIV DNA levels, possibly due to enhanced CD4+ T-cell susceptibility and impaired immune responses [21]. Panobinostat showed no measurable effects on the latent reservoir, consistent with its minimal influence on HIV replication in vitro. These findings highlight the need to critically evaluate the long-term consequences of HDACi in HIV-1 patients [22]. Although these agents show theoretical potential, their inability to substantially reduce the reservoir in vivo underscores the complexity of latency and the necessity for alternative therapeutic strategies to effectively target persistent HIV infection.

An exception to this is a recent study employing the B-cell lymphoma 2 (BCL-2) antagonist, ABT-199, which was based on the finding that BCL-2 overexpression in CD4+ T cells contributes to evasion from cytotoxic T lymphocytes (CTLs) [23]. After treatment with LRAs ex vivo, the combination of CTLs and ABT-199 was shown to reduce the HIV-1 reservoir. The feasibility of translating this success to in vivo reservoir reductions remains to be explored.

The immune-mediated clearance of HIV can occur in the central nervous system (CNS). This mechanism is hypothesized to play a role in eliminating reactivated cells following latency reversal [24]. HIV-induced inflammatory responses in the brain lead to microglial activation, increased expression of cytokines and chemokines, and the infiltration of peripheral cells such as monocytes and lymphocytes—a process known as viral encephalitis [25]. This inflammatory cascade can result in neuronal damage and degeneration. To prevent such detrimental effects, immune responses in the CNS are tightly regulated [26]. Consequently, the immune-mediated clearance of HIV in the CNS is restricted, potentially limiting the “kill” aspect of eradication strategies.

The inherent resistance of HIV-infected CNS cells to apoptosis, coupled with the constrained capacity for immune-mediated killing, suggests that the effectiveness of the “kill” phase following latency reversal in the CNS is likely to be suboptimal [25]. Furthermore, if LRAs induce viral reactivation and immune activation without effective clearance of reactivated cells, this could exacerbate neuronal injury and impair cognitive function. Therefore, the utility of employing LRAs to trigger immune activation in the CNS must be critically evaluated, given the potential risks of neuronal damage [25]. The balance between therapeutic benefits and the risk of neurotoxicity remains a critical consideration when treating virally suppressed individuals with HIV.

Longitudinal assessments rely on viral RNA levels and biomarkers of neuroinflammation and neuronal injury in cerebrospinal fluid. Although this approach is valuable, it is limited due to the invasive nature of lumbar punctures [27]. Less invasive alternatives, such as imaging techniques, have emerged as potential tools for studying CNS effects in vivo. Magnetic resonance imaging, while widely used, does not provide cellular-level resolution or detect changes such as neuronal death. More informative techniques include nuclear imaging approaches, such as single-photon emission computed tomography and positron emission tomography. These modalities use radioactive tracers to detect immune activation, inflammation, and neuronal injury associated with HIV infection. For instance, positron emission tomography imaging of macrophage-colony-stimulating factor 1 receptors has been shown to track microglial neuroinflammation, and tracers for monitoring synaptic density are available [27]. However, to directly visualize HIV-infected cells in the CNS, the development of HIV-specific tracers capable of penetrating the blood–brain barrier is urgently needed. Another promising technique is metabolic imaging via magnetic resonance spectroscopy, which measures chemical changes in neurometabolites, allowing the monitoring of neuroinflammation and associated neuronal injury. However, magnetic resonance spectroscopy is limited to a small number of brain regions, and its utility can be confounded by comorbidities. Overall, imaging techniques hold great promise for evaluating the effects and safety of HIV eradication strategies on the CNS [27]. Despite their limitations in directly monitoring HIV reactivation, the continued development and application of these advanced imaging modalities remain critical for advancing our understanding of CNS-related HIV pathology and the impact of eradication strategies.

### 2.2. Block-and-Lock Strategy Approach

The “block-and-lock” technique is an emerging strategy proposed as a functional cure for HIV-1. This strategy aims to inhibit viral transcription and lock HIV-1 into a latent state, thereby preventing reactivation of the latent reservoir. Its objective is to enable individuals living with HIV-1 to achieve sustained viral remission without requiring ART. This approach mirrors the natural control observed in elite controllers, who maintain high CD4+ T-cell counts, low viral loads, and robust immune function [28]. It can indeed be further elaborated by delving into the molecular mechanisms of transcriptional silencing and this topic, which will be addressed later, will include examples from scientific studies. This strategy involves the use of specific molecules or genetic modifications to lock the virus in a transcriptionally silent state within the host cell. At the molecular level, it typically targets key transcription factors or epigenetic regulators that maintain HIV latency. By enhancing the understanding of how these molecular mechanisms, such as chromatin remodeling, DNA methylation, and histone modifications, contribute to silencing viral gene expression, researchers can identify more precise targets to sustain the latent state and prevent viral reactivation. This could lead to improved therapeutic strategies for HIV eradication.

The underlying premise of the block-and-lock strategy is that HIV-1 can be stably suppressed if its genome and chromatin are permanently silenced, obviating the need for continuous ART. Rather than eliminating the virus, the strategy aims to “freeze” it in a transcriptionally silent state, effectively achieving a functional cure. One key mechanism involves the transcriptional silencing of proviruses [29].

Small interfering RNAs and short hairpin RNAs are promising tools in this context. These molecules target conserved regions of the HIV-1 genome, leading to gene silencing [30]. For example, in the J-Lat 9.2 cell model of HIV-1 latency, short hairpin RNAs such as PromA, 143, and PromA/143 demonstrated the ability to maintain viral inactivity even in the presence of LRAs, while keeping the chromatin compacted [31]. This suggests that short hairpin RNAs could confer resistance to provirus reactivation. Similarly, small interfering RNAs si143, which targets tandem NF-κB motifs within the viral 5′LTR, has been shown to induce transcriptional silencing [32]. Small interfering RNAs and short hairpin RNAs hold considerable promise due to their specificity, potency, and adaptability—these are critical advantages given HIV-1’s high mutational rate [33].

A phase Ib/IIa proof-of-concept trial investigated whether romidepsin, a histone deacetylase inhibitor, could reverse HIV-1 latency in humans while maintaining ART [31]. Six aviremic HIV-1-infected adults received weekly intravenous doses of 5 mg/m^2^ romidepsin for three weeks. The treatment significantly increased histone H3 acetylation (3.7–7.7-fold) and HIV-1 transcription (2.4–5.0-fold, *p* = 0.03), as measured by unspliced cell-associated HIV-1 RNA. Plasma HIV-1 RNA levels, initially <20 copies/mL, rose to quantifiable levels (46–103 copies/mL, *p* = 0.04) in five participants after the second infusion. Importantly, romidepsin did not reduce HIV-specific T-cell counts or impair their cytokine production. Adverse events were mild (grades 1–2) and aligned with known romidepsin side effects [34,35]. These findings demonstrate that significant HIV-1 latency reversal is achievable in vivo without compromising T-cell-mediated immune responses, marking an important step toward strategies targeting the HIV-1 reservoir.

Another promising molecule in the block-and-lock strategy is didehydro-cortistatin A, a potent inhibitor of the HIV-1 Tat protein, whose unique structural modifications enhance its selectivity and potency as a Tat inhibitor. Didehydro-cortistatin A exhibits a high affinity for the transactivation-responsive RNA-binding region of Tat and disrupts Tat-mediated transcriptional feedback loops, effectively “freezing” proviruses into a long-lived, inactive state [34,35]. Remarkably, didehydro-cortistatin A remains effective even after the cessation of treatment. Its superior pharmacokinetic profile, stability, and ability to cross the blood–brain barrier further distinguish it as a promising candidate for targeting viral reservoirs, particularly in the CNS. These characteristics underscore its unique potential for clinical applications in HIV-1 eradication strategies. Remarkably, didehydro-cortistatin A remains effective even after the cessation of treatment [35]. In vivo studies using a bone marrow–liver–thymus mouse model demonstrated that didehydro-cortistatin A suppressed HIV expression by inducing epigenetic silencing via restricted recruitment of RNA polymerase II to the promoter [36]. Didehydro-cortistatin A also alters Tat’s protein environment, enhancing its resistance to proteolytic degradation and interfering with the Tat–TAR interaction, which is essential for Tat functionality. Consequently, didehydro-cortistatin A prevents proviral reactivation [36].

LEDGINs are other promising class of small molecules that target the LEDGF/p75 binding pocket on HIV-1 integrase. LEDGINs inhibit the interaction between LEDGF/p75 and integrase, which has been shown in vitro to increase the fraction of integrated provirus with a transcriptionally silent phenotype. For LEDGINs, their efficacy is contingent on early administration, as their therapeutic window is limited [37]. A study employed a single-cell branched DNA imaging technique to simultaneously detect viral DNA and RNA, allowing for a detailed assessment of the impact of LEDGIN treatment on HIV-1 integration, transcription, and reactivation in both cell lines and primary cells. These findings demonstrated that LEDGIN-mediated retargeting reduces basal transcriptional activity and impairs proviral reactivation, as evidenced by a significant decrease in viral RNA expression per residual provirus. The interaction between HIV-1 integrase and the chromatin tethering factor LEDGF/p75 serves as a key determinant of integration site preference [37]. By employing LEDGINs to disrupt this interaction, researchers have elucidated how integration retargeting influences the three-dimensional positioning of the provirus, its transcriptional activity, and its susceptibility to reactivation [37]. These results support the feasibility of a “block-and-lock” strategy aimed at permanently silencing HIV-1 by directing integration into genomic regions refractory to reactivation following ART discontinuation.

Moreover, treatment of primary cells with LEDGINs resulted in an enrichment of proviruses in a deep latency state. These data reinforce the critical role of integration site selection in dictating HIV-1 transcriptional fate and provide compelling evidence for “block-and-lock” functional cure strategies, wherein HIV-1 reservoirs are permanently silenced through integration retargeting.

Tat-mediated transcriptional activation can also be suppressed by targeting splicing factor 3B subunit 1, a critical HIV dependency factor. Experimental evidence suggests that inhibiting splicing factor 3B subunit 1 prevents viral reactivation, even in the presence of various reversal agents [38].

Another key target in the block-and-lock strategy is the mammalian target of rapamycinm (mTOR), a conserved serine/threonine kinase complex that regulates HIV-1 latency. Compounds such as Torin1 and pp242 have been shown to inhibit mTOR activity, thereby suppressing HIV transcription by interfering with Tat-dependent mechanisms in vivo [39].

Beyond transcriptional silencing, post-transcriptional gene silencing represents an additional mechanism to achieve the block and lock of latent reservoirs. Small molecule modulators involved in post-transcriptional regulation of HIV-1 can be inhibited to suppress viral expression. For instance, RNA surveillance proteins UPF2 and SMG6 interact with UPF1 to mediate post-transcriptional silencing, significantly reducing HIV-1 gene expression in infected primary CD4+ T cells [40]. Synthetic siRNAs targeting conserved regions within the V3 loop and the CD4 binding site of gp120 have also demonstrated efficacy in silencing HIV-1 gene expression [28].

In summary, the block-and-lock strategy aims to establish a transcriptionally and post-transcriptionally silent state in the HIV-1 reservoir, enabling long-term viral remission without ART. While still in its early stages, the strategy holds significant potential as a functional cure for HIV-1.

### 2.3. Broadly Neutralizing Antibodies Strategy

Another approach to reducing the HIV-1 proviral reservoir involves leveraging bNAbs that target HIV-1. This strategy aims to confer resistance to infection, effectively mimicking a vaccine-like effect by neutralizing the virus during acute infection or after reactivation of latent reservoirs. Progress has been made in identifying bNAbs capable of neutralizing both cell-free and cell-associated HIV-1 infection [4]. In vitro studies have demonstrated the ability of bNAbs to limit viral replication [41], while research in a rhesus macaque simian/human immunodeficiency virus model has shown that bNAbs can achieve transient viral suppression in chronic infections [42]. Importantly, early administration of bNAbs during acute infection appears to extend the duration of viral suppression, indicating a potential role for bNAbs in early-treatment regimens [41]. Additional success has been observed in humanized mouse models, where delivery of bNAbs via adenovirus serotype 5 (Ad5) or adeno-associated virus serotype 1 (AAV1) vectors demonstrated efficacy in controlling viral replication [43]. However, due to the limited half-life of antibodies, an effective and durable delivery system remains essential for sustained therapeutic application.

Despite encouraging results, several limitations complicate the use of bNAbs as a strategy against HIV-1. For instance, while bNAbs can inhibit cell-free infection, their ability to prevent cell-to-cell transmission is only partial, and the underlying mechanisms for this resistance are not yet understood [44]. Furthermore, while transient viral suppression serves as a proof of concept for bNAb efficacy, the finite systemic half-life of bNAbs restricts their utility in chronic HIV-1 infections unless this limitation is addressed [42]. Another challenge is the potential for an anti-antibody immune response, as observed in rhesus macaques treated with AAV-delivered bNAbs, where 17 out of 20 monkeys developed such responses. This study, however, did not evaluate the impact of these immune reactions on viral control [45]. Other studies in Phase 1 have also evaluated additional bNAbs such as PGT121, which targets the V3 loop base, and VRC07-523LS, with results from these studies expected shortly [46].

Combination therapies using bNAbs targeting non-overlapping epitopes, such as 3BNC117 and 10–1074, have shown promise in extending antiviral coverage. Among viremic participants receiving one or three infusions of this combination, those with antibody-sensitive viruses experienced greater declines in viremia compared to monotherapy, with reductions sustained for up to three months. Complete suppression was observed only in participants with low baseline viral loads, and resistance to 3BNC117 was not observed despite persistent viremia and viral recombination. However, the shorter half-life of 3BNC117 resulted in a transition to 10–1074 monotherapy, coinciding with the emergence of 10–1074-resistant variants [47].

In ART-suppressed individuals harboring bNAb-sensitive viruses, the combination of 3BNC117 and 10–1074 effectively maintained suppression during ATI. The median rebound time for seven out of nine participants was 21 weeks (range: 15–26 weeks), with two participants maintaining suppression beyond 30 weeks [47]. Differences in rebound antibody concentrations reflected 10–1074’s longer half-life, but no double-resistant variants emerged. These results underscore the combination’s potential for extended suppression, pending optimization of dosing regimens and exploration of long-acting variants.

Future studies are investigating combinations targeting three distinct envelope epitopes, such as PGT121 with PDGM1400 and VRC07-523LS, or SAR441236, an engineered antibody targeting CD4bs, the V2 loop, and MPER. Engineered antibodies offer potential advantages, including lower costs and streamlined development pathways compared to single-specificity combinations. Targeting non-overlapping epitopes in HIV-1 therapy enhances efficacy and reduces resistance by preventing viral escape through multiple simultaneous mutations, which is often detrimental to viral fitness and replication. This makes it significantly harder for HIV-1 to escape immune surveillance [48]. This approach engages different immune mechanisms, including bNAbs and cytotoxic T cells, leading to synergistic neutralization and improved viral clearance. This multi-pronged attack increases therapeutic potency. Additionally, it provides broader coverage against diverse HIV-1 strains, reducing the risk of treatment failure due to strain variability, and ensures a more durable antiviral response. By making it harder for the virus to evade immune detection, this strategy strengthens long-term viral suppression and supports functional cure efforts [48]. However, their safety and risk of anti-antibody responses remain under evaluation. Results from these ongoing studies will further define the therapeutic potential of multi-specific and combination bNAb therapies against HIV-1 [48].

Also, the HIV-1 fusion peptide is a promising target for vaccine development; however, the global sequence diversity of fusion peptide among circulating strains has restricted the neutralization breadth of anti-fusion peptide antibodies to approximately 60%. In this study, it is utilized in vitro evolution of the fusion-peptide-targeting antibody VRC34.01 to improve its neutralization capacity through site-saturation mutagenesis and yeast display [49]. Iterative rounds of directed evolution, involving selection for binding to resistant HIV-1 strains, identified a variant, VRC34.01_mm28, as a superior antibody with a 10-fold increase in potency relative to the original antibody and an ~80% neutralization breadth against a cross-clade panel of 208 HIV-1 strains. Structural characterization revealed that the optimized paratope expands the fusion-peptide-binding groove, enabling recognition of diverse fusion peptide sequences with varying lengths while simultaneously interacting with the HIV-1 Env backbone. These findings highlight key antibody features necessary for improved breadth and potency against the fusion peptide vulnerability site and support the advancement of broad HIV-1 fusion peptide-targeting vaccines and therapeutic strategies [49].

Lastly, the extensive genetic and antigenic diversity of HIV-1 may limit the broad applicability of bNAbs across different viral strains [48]. Even if these challenges are overcome, bNAbs alone will not directly eliminate proviral DNA from the latent reservoir, underscoring the need for complementary strategies to achieve a cure.

Overall, these findings suggest that while bNAbs exhibit substantial antiviral activity, their monotherapy utility is limited by the emergence of resistant variants, akin to small-molecule antiretroviral drugs.

### 2.4. Transplantation of Hematopoietic Stem Cell for HIV-1 Remission

HIV-1 infection requires the presence of a CD4 receptor and a chemokine coreceptor, primarily CCR5. Homozygosity for a 32-base pair deletion in the CCR5 allele (CCR5Δ32/Δ32) confers resistance to HIV variants utilizing the CCR5 coreceptor for entry [29]. In hematopoietic stem cell transplantation strategies, the aim is to use donor cells lacking the CCR5 receptor, thereby obstructing viral entry into host cells. Donor–host compatibility is determined based on the human leukocyte antigen system, specifically matching five genetic loci (A, B, C, DRB, and DRQ), each with two allelic variants (totaling 10 alleles) [29].

To date, only two successful cases of HIV remission using this approach have been reported. The “London patient”—a Caucasian male with HIV-1 and Hodgkin’s lymphoma—underwent allogeneic HSCT using cells from a CCR5Δ32/Δ32 donor, achieving sustained remission [50,51]. This case mirrors the earlier success of the “Berlin patient”, who also achieved HIV remission following HSCT from a CCR5Δ32/Δ32 donor. The Berlin patient, who had acute myeloid leukemia and HIV-1, remained in remission for 20 months post-transplantation and after discontinuing ART.

A key distinction between these cases is that the London patient achieved remission after a single transplantation using reduced-intensity drug regimens and without requiring total body irradiation. In contrast, the Berlin patient experienced cancer relapse, necessitating additional chemotherapy and a second transplantation before achieving remission [50]. At 28 months post-treatment, the London patient showed no signs of viral rebound, with undetectable viral load in semen and negative HIV-1 DNA results in tissue samples from the rectum, caecum, sigmoid colon, and terminal ileum. However, lymph-node tissue was positive for HIV long terminal repeats, indicating the presence of viral remnants incapable of replication [51].

The absence of viral rebound after ART interruption in these cases has been attributed to a significant reduction in the latent reservoir and the near-complete replacement of host CCR5-expressing cells, which serve as targets for HIV [52]. These findings demonstrate that sustained HIV remission is achievable through successful HSCT. However, other cases involving bone marrow transplantation a donor encoding the CCR5Δ32/Δ32 mutation have resulted in viral rebound within weeks of ART interruption, despite notable reductions in the HIV reservoir [53].

While promising, this strategy is associated with significant risks and limitations. In the context of medical conditions, the primary challenges include disease relapse, infectious complications, and regimen-related toxicities, particularly among older patient populations [54]. Careful consideration must be given to quality of life and long-term outcomes, especially for elderly individuals. Access to HSCT is influenced by socioeconomic status, education level, and ethnicity, with migrants and minority groups frequently encountering systemic barriers [54]. Additionally, cultural beliefs and language barriers may hinder patient comprehension and adherence to HSCT protocols. Immigrant populations often exhibit distinct perceptions of illness and treatment, which can impact their interaction with and utilization of healthcare systems [54].

HSCT raises significant ethical concerns, particularly in the domains of informed consent, donor–recipient matching, and the long-term implications of transplantation. The decision-making process for HSCT is inherently complex, involving multiple stakeholders, including patients, families, and healthcare providers [55]. This process necessitates a careful balance between potential risks, clinical benefits, and the overall impact on the patient’s quality of life. Pediatric HSCT introduces additional ethical challenges, particularly concerning the involvement of minors in decision-making processes and the long-term physical, psychological, and social effects on survivors. A particularly contentious issue is the right of mature minors to refuse life-sustaining treatments, such as HSCT, which often intersects with legal, ethical, and familial considerations. Furthermore, conflicts of interest in HSCT research and clinical practice may arise from various sources, including economic pressures, selective publication of data, and the prioritization of research that is financially lucrative over studies addressing unmet medical needs [55]. These conflicts can undermine the integrity of research and the equitable delivery of care. Ethical oversight committees play a pivotal role in addressing these challenges by safeguarding the rights and safety of research participants. These committees are responsible for ensuring compliance with internationally recognized ethical guidelines, such as the Declaration of Helsinki [55]. To maintain their credibility and effectiveness, ethical committees must operate independently and ensure that the benefits and risks of research are equitably distributed across all societal groups, thereby promoting justice and inclusivity in scientific advancement.

HSCT is a highly invasive and morbid procedure, often associated with graft-versus-host disease and challenges in finding HLA-compatible CCR5Δ32/Δ32 donors [46]. Furthermore, the strategy does not protect against HIV variants utilizing the CXCR4 coreceptor for cell entry, leaving the potential for continued viral replication [56]. These challenges underscore the need for careful consideration and further research to improve the safety and feasibility of this approach.

Patients infected with CXCR4-tropic HIV generally exhibit a poorer clinical prognosis and would not benefit from the transplantation of CCR5Δ32 hematopoietic stem and progenitor cells, as the CXCR4-tropic virus can independently infect cells without relying on the CCR5 coreceptor [57]. There is a critical need for strategies to combat the multiple variants of HIV that evolve in each patient, as well as the identification of therapies effective against both CCR5- and CXCR4-tropic HIV-1. Developing diverse genetic resistance mechanisms is comparable to the requirement for maintaining multiple small-molecule inhibitors during ART to control viral replication [57]. The development of an autologous hematopoietic stem cell therapy could improve transplantation safety, enhance treatment efficacy by providing resistance to both dual- and CXCR4-tropic HIV, and expand the pool of HIV patients eligible for HSCT.

CRISPR/Cas9 technologies—including base editing, prime editing, and zinc-finger nucleases—have emerged as promising approaches with distinct advantages in specificity, efficiency, and reduced off-target effects. These methods could provide solutions to challenges associated with conventional gene therapy, such as immune responses, delivery efficiency, and long-term genomic stability.

The diverse applications of this technology, including in vivo, in vitro, and ex vivo approaches, highlight its versatility in medical research and in evaluating key limitations that may impact its future clinical translation.

### 2.5. Cell-Based Immunotherapies to Eliminate HIV-Infected Cells

Cell-based immunotherapies for HIV-1 are designed to leverage and amplify the body’s natural immune mechanisms for identifying and eliminating HIV-infected cells. These approaches generally involve the manipulation and administration of immune cells, such as cytotoxic T lymphocytes, natural killer cells, and dendritic cells. Among the most extensively investigated strategies are the adoptive transfer of ex vivo-expanded HIV-specific cytotoxic T lymphocytes and the engineering of CAR T cells that are tailored to recognize and eliminate HIV-infected cells [58]. The fundamental principle of CAR T-cell therapy for HIV-1 treatment involves the ex vivo transduction of patient-derived CD8+ T cells with a CAR construct. Following transduction, these modified cells are expanded and reinfused into the patient, where the CAR T cells specifically target and eliminate HIV-1-infected cells, thereby reducing the reservoir of infected cells [59]. This approach has demonstrated experimental success. A novel two-molecule CAR strategy, termed duoCAR, introduced via a single lentiviral vector, has exhibited remarkable efficacy in a humanized NOD scid gamma mouse model. The duoCAR design incorporates two Env-binding domains, targeting distinct gp120 epitopes whose accessibility varies with conformational changes following CD4 and gp120 interaction [60]. This approach reduced the population of HIV-1-infected cells by over 97% in vivo and demonstrated effectiveness against bNAb-resistant strains of HIV-1 [61].

Preclinical studies and early-phase clinical trials have demonstrated the potential of ex vivo-expanded HIV-specific immune cytotoxic T lymphocytes in controlling viral replication and slowing disease progression [62]. For instance, trials involving the infusion of autologous HIV-specific cytotoxic T lymphocytes in patients receiving antiretroviral therapy have shown that these cells can persist in vivo and exhibit antiviral activity. These results highlight the potential role of cytotoxic T lymphocytes in maintaining viral suppression and contributing to the immune control of HIV-1 [63].

In these trials, CCR5 is a critical coreceptor for HIV entry. This study, for example, assessed the safety of gene editing using zinc-finger nucleases to permanently disrupt the CCR5 gene in autologous CD4 T cells, followed by their infusion into patients with chronic aviremic HIV. In this case, twelve patients received a single infusion of 10 billion autologous CD4 T cells, 11–28% of which were genetically modified with zinc-finger nucleases. Six patients underwent a treatment interruption of ART four weeks post-infusion. Safety, immune reconstitution, and HIV resistance were evaluated. As for the results, there was one serious adverse event that occurred, attributed to a transfusion reaction. Median CD4 T-cell counts rose significantly from 448 to 1517 cells/mm^3^ at week 1 (*p* < 0.001). CCR5-modified cells represented 8.8% of peripheral mononuclear cells and 13.9% of CD4 T cells, with an estimated half-life of 48 weeks. During ART interruption, CCR5-modified cells declined more slowly than unmodified cells (−1.81 vs. −7.25 cells/day, *p* = 0.02). HIV RNA became undetectable in one of four evaluable patients, and most showed reduced HIV DNA levels. Infusions of CCR5-modified CD4 T cells appear safe and show potential for immune reconstitution and HIV control [63].

Despite these advancements, several significant challenges remain in the development of CAR T-cell therapies for HIV-1 infection. One major concern is the potential for severe cytokine release syndrome associated with CAR T-cell treatment [59]. Furthermore, CAR T cells have shown limited efficacy in vitro against follicular dendritic cells harboring surface-bound HIV-1, a key component of the latent viral reservoir [61]. Recent studies suggest that increasing dendritic cells recruitment and function could enhance T-cell responses and improve therapeutic efficacy. Combination strategies that integrate dendritic cell vaccines, chimeric antigen receptor CAR T cells, and immune checkpoint inhibitors have demonstrated promise in preclinical and clinical studies. Notably, glioblastoma patients treated with dendritic cell vaccines in conjunction with Treg depletion, anti-PD-1 therapy, and adjuvants such as Poly I:C have shown extended progression-free survival [62]. These findings underscore the necessity of combination therapies to overcome glioblastoma’s immune barriers. Therefore, targeting follicular dendritic cells should be approached through strategies that not only enhance their antigen-presenting function but also promote a favorable immune microenvironment. This may involve optimizing CAR designs to engage both DCs and T cells, utilizing adjuvants that support dendritic cell maturation and activation, and incorporating checkpoint blockade to counteract immunosuppression [62]. A multifaceted approach that restores the dendritic cell–T cell axis will likely be essential to achieving durable immune responses in glioblastoma treatment [62].

Issues with CAR T-cell persistence and lack of robust expansion have also been reported [63]. Another critical challenge is the risk of off-target effects, as evidenced by B-cell aplasia observed in leukemia studies targeting the B-cell antigen CD19 with CAR T cells [64].

Addressing these limitations is essential in realizing the full therapeutic potential of CAR T cells in HIV-1 treatment. If these obstacles can be overcome, CAR T-cell therapies hold significant promise as a transformative approach to combating HIV-1. Ongoing research focuses on optimizing processes for immune cell selection and expansion, leveraging cytokines and growth factors to enhance cellular proliferation and functional capacity [58]. Another major hurdle is ensuring the in vivo persistence and proper trafficking of infused cells. For therapeutic efficacy, these cells must survive and localize to sites of HIV replication, such as lymphoid tissues and viral reservoirs. Strategies to address this include the genetic modification of T cells to express anti-apoptotic genes and the use of adjunctive therapies to improve cell survival and function [58]. Additionally, advances in imaging technologies are providing critical insights into the distribution and longevity of infused cells, aiding in the refinement of therapeutic protocols. In conclusion, cell-based immunotherapies represent a transformative approach to HIV-1 treatment, with the potential to enhance antiviral immunity and selectively target HIV-infected cells. While substantial challenges remain, ongoing research and technological progress are steadily addressing these obstacles. Continued exploration of novel strategies, combined with rigorous clinical evaluation, will be essential for fully realizing the therapeutic potential of these approaches in combating HIV-1.

### 2.6. Clustered Regularly Interspaced Short Palindromic Repeats Strategy

CRISPR method and the associated CRISPR-associated protein 9 (Cas9) represent a sophisticated genome-editing technology that employs RNA to specifically target DNA sequences [65]. This technology offers two principal strategies for combating HIV infection: targeting host genes or directly targeting the viral genome [66]. The use of CRISPR/Cas9 to target HIV-1 proviral DNA within the latent reservoir represents a unique strategy that differs fundamentally from coreceptor gene disruption. Despite existing challenges that must be addressed to advance this technology into a practical therapeutic option, considerable progress has been made [66].

In the context of host gene targeting, CRISPR/Cas9 has been used to introduce deletions in both alleles of the CCR5 chemokine receptor in induced pluripotent stem cells. Additionally, the CXCR4 receptor has been effectively targeted using this method [66].

The application of CRISPR/Cas9 technology to HIV-1/AIDS treatment was first demonstrated in 2013 by Ebina et al. In their study, CRISPR/Cas9 was utilized to suppress HIV-1 gene expression in Jurkat cell lines by targeting two critical regions within the HIV-1 long terminal repeat: the NF-κB binding motifs in the U3 region and the TAR sequences within the R region [67]. This approach led to a significant inhibition of HIV-1 proviral transcription and replication. Notably, the study provided evidence that CRISPR/Cas9 could eliminate integrated viral sequences from host cell genomes, highlighting its potential as a novel therapeutic strategy against HIV-1 [67]. Following this, in 2014, Hu et al. investigated the excision of HIV-1 sequences using Cas9 guided by RNA (gRNA) to target conserved sites within the U3 region of the LTR [68]. Their results demonstrated successful suppression of viral gene activity and reduced viral replication in latent HIV-1-infected T cells, pro-monocytic cells, and microglial cells. Importantly, the study reported minimal genotoxicity and no detectable off-target effects, underscoring the precision of the CRISPR/Cas9 system [65]. CRISPR/Cas9 has also been utilized to reactivate viral gene expression in HIV-1 latency models [69]. Liao et al. (2015) showed that CRISPR/Cas9 could disrupt the genomes of latent, HIV-1-infected cells and confer resistance against new infections when incorporated into pluripotent stem cells, which maintained resistance following differentiation [68]. This finding was further corroborated by studies demonstrating enhanced transcription of latent HIV-1 when CRISPR/Cas9 was combined with LRAs [70]. More recently, Wang and col. employed a *Staphylococcus aureus* Cas9 (SaCas9) system, combined with gRNAs in a single lentiviral vector, to excise latent HIV-1 provirus and suppress reactivation. The study also revealed that using multiple gRNAs in combination significantly enhanced the disruption of the HIV-1 genome compared to single gRNA-guided SaCas9 editing, demonstrating the superiority of a multiplexed approach [71].

CRISPR/Cas9-mediated mutational inactivation of the HIV-1 provirus using a single sgRNA has been demonstrated [72]. By targeting key viral sequences within the LTR and critical replication genes, this approach effectively introduced mutations that inactivated the HIV-1 provirus. However, single sgRNA strategies are vulnerable to viral escape due to incomplete cleavage. To overcome this limitation, combinatorial CRISPR/Cas9 strategies employing multiple sgRNAs have been proposed to enhance gene-editing efficacy and reduce escape potential [73]. In a study, CRISPR/Cas9 was applied to target the LTR in latently infected T cell lines. Sequencing data revealed site-specific mutations that suppressed viral gene expression and production, even under stimulation with tumor necrosis factor alpha (TNFα). Additional studies explored the targeting of incoming HIV-1 using CRISPR/Cas9 systems, demonstrating that dual sgRNAs achieve superior cleavage efficiency compared to single sgRNAs, particularly in non-integrated HIV-1 reporter plasmids [74,75]. Interestingly, recent findings suggest that CRISPR/Cas9 can also cleave non-integrated HIV-1 DNA, resulting in a significant 3–4-fold reduction in integrated proviral DNA. Notably, the non-homologous end-joining DNA repair pathway was implicated in repairing cleavage sites within non-integrated HIV-1 DNA [74]. These findings highlight the ability of CRISPR/Cas9 to target both integrated proviral DNA and non-integrated viral genomes in latently infected cells, reinforcing its potential as a therapeutic tool for HIV-1/AIDS treatment.

The feasibility of HIV-1 proviral DNA excision was further validated in animal models [75]. Researchers utilized an AAV-based delivery system encoding SaCas9 and multiplex sgRNAs to disrupt HIV-1 provirus in three distinct animal models. In Tg26 mice, the intravenous administration of quadruplex sgRNA/SaCas9 AAV-DJ/8 efficiently cleaved proviral DNA, leading to a marked reduction in viral replication. Similarly, intravenous delivery of the same system in humanized bone marrow–liver–thymus (BLT) mice infected with HIV-1 enabled proviral cleavage across multiple tissues, including the brain, colon, spleen, heart, and lungs. Furthermore, this approach successfully suppressed HIV-1 replication in Eco-HIV acutely infected mice [76]. The successful excision of HIV-1 proviral DNA using SaCas9 and sgRNA in vivo, facilitated through AAV delivery, represents a significant milestone and paves the way for future clinical trials aimed at curing HIV-1 infection in humans. Also, the CRISPR/Cas9 system has been extensively utilized to disrupt CCR5 and CXCR4, two key coreceptors involved in HIV-1 infection. In 2013, it was successfully demonstrated that CCR5 silencing could be achieved in human embryonic kidney 293T cells via transfection with Cas9 and specific sgRNAs [77]. Building on this work, Ye and colleagues combined CRISPR/Cas9 or TALENs with piggyBac technology to introduce a homozygous CCR5Δ32 mutation in induced pluripotent stem cells. These modified induced pluripotent stem cells were capable of differentiating into monocytes/macrophages resistant to HIV-1 infection, paving the way for novel therapeutic applications [78]. Another study employed adenoviral delivery of CRISPR/Cas9 alongside sgRNAs targeting exon 4 of the CCR5 gene, achieving over 60% editing efficiency in TZM-BL cells. These findings were further validated in Chinese hamster ovary cells and human T-cell lines. By using a chimeric Ad5/F35 adenoviral vector, they successfully delivered the CRISPR/Cas9 system to human CD4+ T cells, efficiently silencing CCR5 expression and protecting these cells against HIV-1 infection with minimal off-target effects [79]. Later, CRISPR/Cas9 has been utilized to disrupt CCR5 in human CD34+ hematopoietic stem and progenitor cells, achieving durable CCR5 knockout in vivo. This modification effectively inhibited HIV-1 infection and showed stable CCR5 disruption in secondary repopulating hematopoietic stem cells, offering a promising foundation for future HIV-1 therapies involving CCR5-edited HSC transplantation [80].

Simultaneously, advancements were made in disrupting CXCR4 expression using CRISPR/Cas9. Hou et al. demonstrated efficient disruption of CXCR4 in both human and rhesus macaque CD4+ T cells by delivering CRISPR/Cas9 with lentiviral vectors and two sgRNAs targeting conserved CXCR4 sequences. The edited human CD4+ T cells showed resistance to X4-tropic HIV-1 infection, as evidenced by reduced p24 levels, with no detectable off-target effects or toxicity [81]. Similarly, Schumann et al. employed Cas9 ribonucleoprotein complexes to disrupt CXCR4 expression in human CD4+ T cells, achieving approximately 40% reduction in CXCR4 surface expression. When paired with a repair template, targeted modifications were introduced with knock-in efficiencies reaching up to 20%, as confirmed by deep sequencing [82].

Moreover, Liu et al. successfully generated CXCR4 P191A mutants with anti-HIV-1 properties while preserving CXCR4 functionality using CRISPR/Cas9 and piggyBac technologies. This suggests that targeting CXCR4 in mature post-thymic CD4+ T cells could serve as a viable strategy for HIV-1 therapy without impairing critical CXCR4 functions [83].

In a study involving HIV-1ADA-infected CD34+ NSG-humanized mice, treatment with long-acting ester prodrugs of cabotegravir, lamivudine, and abacavir, combined with native rilpivirine, was followed by dual CRISPR-Cas9 gene editing, targeting the CCR5 and HIV-1 proviral DNA. This sequential approach achieved viral suppression, restoration of absolute human CD4+ T-cell counts, and elimination of replication-competent virus in 58% of the infected mice. Dual CRISPR therapy facilitated the excision of integrated proviral DNA in infected human cells within live animals. Advanced techniques, including highly sensitive nested and droplet digital polymerase chain reaction, RNAscope, and viral outgrowth assays, confirmed the complete elimination of HIV-1. No viral presence was detected in the blood, spleen, lung, kidney, liver, gut, bone marrow, or brain tissues of virus-free animals. Additionally, progeny virus was neither detected nor recoverable from adoptively transferred CRISPR-treated virus-free mice. In contrast, untreated or viral-rebounded animals exhibited residual HIV-1 DNA fragments. Importantly, no off-target toxicities were observed in any of the animals treated. Statistical analysis revealed that dual CRISPR therapy significantly outperformed single treatments in achieving HIV-1 elimination. Collectively, these findings highlight the critical potential of combinatorial CRISPR-Cas9 gene editing as a transformative approach for the eradication of HIV-1 infection [84]. These off-target genome editing approaches refer to unintended genetic modifications caused by engineered nucleases, which can still occur in DNA sequences with minor mismatches in the protospacer-adjacent motif (PAM)–distal region of the sgRNA [84]. These off-target sites contain noncanonical PAMs and diverse nucleotide variations, with mismatches being more tolerated at the 5′ end than the 3′ end of the gRNA [85]. A mismatch in the seed region can hinder Cas9 activation, while three or more mismatches can disrupt HNH conformation, preventing cleavage. The structural properties of gRNAs significantly influence off-target effects, which not only reduce CRISPR-Cas9’s therapeutic precision but also compromise gene function studies [86]. These effects may lead to adverse consequences such as unintended DNA damage, immune activation, and cytotoxicity.

Recent strategies to mitigate CRISPR-Cas9’s off-target effects focus on optimizing sgRNA design, engineering Cas variants, and employing novel editing systems and inhibitors:

-Improving sgRNA specificity: Adjustments to sgRNA, such as optimizing GC content (40–60%), shortening sgRNA sequences (<20 nucleotides), or chemically modifying the sgRNA backbone, have enhanced target specificity [85]. Techniques like the “GG20” method, replacing bases at the sgRNA 5′ end with guanines, further minimize off-target interactions without sacrificing on-target efficiency [87].

-Enhanced Cas variants: Engineered Cas9 mutants like enhanced-specificity Cas9 (eSpCas9) and SpCas9-HF1 exhibit higher fidelity by reducing non-specific DNA interactions while retaining on-target activity. Cas9 nickase variants, which cleave single DNA strands, also reduce collateral damage and improve accuracy. Additionally, Cas9 homologs requiring rare PAM sequences, such as SaCas9, offer precise targeting with reduced off-target potential [88].

-Prime editing: Prime editing eliminates the need for double-strand breaks and donor DNA, significantly lowering off-target risks. Utilizing nicking Cas9 (nCas9), reverse transcriptase, and pegRNA, this system achieves precise base editing with minimal unintended consequences, though its efficiency remains a challenge [89].

-Anti-CRISPR Proteins: Anti-CRISPR proteins inhibit CRISPR-Cas activity by blocking DNA binding or cleavage, enhancing target selectivity, and mitigating off-target effects. Anti-CRISPR proteins provide a safety mechanism for genome editing but require further exploration for broader applications [90].

-SuperFi-Cas9: The recently developed Super-Fidelity Cas9 (SuperFi-Cas9), engineered to disrupt mismatch stabilization, exhibits 4000-fold improved fidelity compared to wild-type Cas9. Early studies demonstrate its high specificity, though its in vivo applications remain limited [91].

These advancements collectively represent significant progress in reducing off-target genome editing effects, paving the way for safer and more precise CRISPR-based therapies.

These findings highlight the diverse potential targets within the HIV-1 genome that can be exploited to reverse latency in infected cells. However, further research is essential to identify the most effective genomic targets and optimize the therapeutic application of CRISPR/Cas9 technology in HIV-1 treatment.

### 2.7. Bispecific Antibodies Method

At low Env densities, interprotein binding of bNAbs—where both arms simultaneously engage two separate Env trimers—is improbable. Consequently, bispecific bNAb designs have concentrated on intraspike binding, enabling both arms to target distinct epitopes within the same Env trimer. This approach enhances bNAb avidity and has demonstrated improved neutralization potency and breadth [92]. Bispecific antibodies (bsAbs) can be categorized into two molecular formats based on the presence or absence of an Fc domain. The IgG format offers advantages such as extended half-life and the ability to elicit Fc-mediated effector functions. Conversely, bispecific non-IgG formats, comprising single-chain variable fragments or nanobodies, are advantageous due to their smaller size [93], which facilitates enhanced tissue penetration and enables binding to epitopes that may be inaccessible to traditional IgG antibodies. Notably, bivalent and bispecific nanobodies have exhibited significant improvements in neutralization potency and antiviral activity compared to equimolar concentrations of their monovalent counterparts [94]. In summary, bispecific antibodies serve as a valuable strategy to increase avidity by concurrently binding two different epitopes on the Env trimer, thereby enhancing their neutralization potency.

Multiple research groups have engineered bispecific antibodies that simultaneously target the HIV-1 envelope glycoprotein (Env) and engage effector cells [95]. This dual targeting facilitates the recruitment of effector cells to HIV-1-infected cells, thereby enhancing the efficiency of cell-mediated cytotoxicity. For instance, Ramadoss and colleagues developed single-chain diabodies that bind both HIV-1 Env and CD16 on natural killer (NK) cells, promoting robust NK cell activation and lysis of infected cells [96]. Despite promising in vitro results, translating these findings into in vivo models has presented challenges. In a study involving simian–human immunodeficiency virus (SHIV)-infected monkeys, the administration of dual-affinity retargeting molecules targeting HIV-1 Env and CD3, in combination with a latency-reversing agent, did not lead to a significant reduction in latently infected cells. Moreover, the treatment induced the production of anti-drug antibodies, resulting in a rapid decline in serum concentration after multiple infusions [97]. Overall, bispecific antibodies offer a promising avenue for HIV-1 cure strategies by facilitating the targeted elimination of infected cells through immune activation. However, further research is necessary to address the challenges observed in in vivo applications and to optimize their therapeutic potential.

Chronic HIV-1 infection precipitates a progressive decline in immune cell functionality, thereby compromising the host’s ability to control viral replication [98]. This deterioration is attributed to sustained exposure to viral antigens, leading to chronic activation of immune cells and culminating in a state of exhaustion. In this exhausted state, CD8+ T cells and NK cells exhibit diminished cytotoxic capabilities, impairing their capacity to eliminate HIV-infected cells [99]. Additionally, acute HIV-1 infection is marked by a significant reduction in peripheral CD4+ T-cell counts. In individuals living with HIV, not only are CD4+ T-cell counts reduced, but these cells also display a decreased ability to secrete cytokines and an increased expression of inhibitory receptors [99]. Given that CD4+ T cells produce supportive cytokines that directly enhance CD8+ T-cell responses, their depletion adversely affects CD8+ T-cell efficacy. This loss of effector cell function poses challenges in developing therapies that depend on the host’s immune system [100]. There is a clinical trial that studied the T-cell dysfunction that develops early during HIV infection, contributing to non-AIDS-related complications and limiting curative potential. Early initiation of ART during primary HIV infection can mitigate this dysfunction, but the degree of immune recovery remains unclear. In a study of 66 individuals treated during early primary HIV infection for up to three years, researchers observed persistent T-cell activation and elevated expression of immune checkpoint receptors such as PD1, Tim-3, and TIGIT, even with ART. While ART partially normalized immune checkpoint receptor expression, the recovery was incomplete and varied by receptor. Epigenetic analysis of HIV-specific CD8 T cells revealed features of exhaustion typically associated with chronic infection present early in primary HIV infection. ART initiated during this stage partially shifted the epigenome toward a memory phenotype, though changes were incomplete. These results indicate that, while early ART provides substantial immune restoration, HIV-associated phenotypic and epigenetic alterations may not be fully resolved [100]. Consequently, strategies such as bispecific antibodies are under investigation to counteract immune cell exhaustion.

**Table 1 life-15-00276-t001:** Recent advancements in the therapeutic strategies for targeting HIV-1 virus.

Study	Year	Method	Main Findings	Reference
Barton et al.	2016	“Shock-and-Kill”	Romidepsin effectively reactivated replication-competent SIV in post-antiretroviral therapy controllers.	[18]
Policicchio et al.	2016		Romidepsin effectively reactivated replication-competent SIV in post-antiretroviral therapy controllers	[19]
Ren et al.	2020		BCL-2 antagonism sensitized resistant reservoirs to cytotoxic T-cell-mediated elimination.	[23]
Mousseau et al.	2015	“Block-and-lock”	Successfully prevented HIV-1 reactivation from latency.	[33]
Kessing et al.	2017		Demonstrated suppression of HIV rebound after treatment interruption.	[34]
Mediouni et al.	2019		Effectively inhibited HIV-1 replication by targeting the Tat region.	[36]
Huang et al.	2016	Broadly neutralizing antibody.	Developed a CD4-binding site antibody with near-pan neutralization breadth.	[41]
Gautam et al.	2018		Achieved durable protection from SHIV infection.	[42]
Badamchi-Zadeh et al.	2018		Demonstrated therapeutic efficacy in HIV-1-infected models.	[43]
Gupta et al.	2019	Hematopoietic stem-cell transplantation.	Reported HIV-1 remission following transplantation.	[50]
Gupta et al.	2020		Evidence for HIV-1 cure after 30 months of treatment interruption.	[51]
Duarte et al.	2015		Documented a case report of transplantation leading to HIV suppression.	[52]
Anthony-Gonda et al.	2019	CAR-T cell therapy	Demonstrated broad in vitro and potent in vivo elimination of HIV-infected cells.	[60]
Ollerton et al.	2020		CAR-T cells failed to recognize and eliminate follicular dendritic cell reservoirs.	[61]
Tebas et al.	2014		Explored safety and efficacy of gene editing for functional HIV cure.	[63]
Ebina et al.	2013	CRISPR/Cas9	Demonstrated potential for using CRISPR to disrupt latent reservoirs.	[67]
Liao et al.	2015		Successfully used CRISPR/Cas9 as a defense mechanism against HIV infection.	[70]
Lebbink et al.	2017		Prevented HIV replication and viral escape using CRISPR.	[73]

## 3. Conclusions and Future Directions

Future research in stem cell therapy for HIV-1 should focus on refining treatment strategies to maximize therapeutic efficacy while minimizing risks and adverse effects. This includes “shock-and-kill” therapies, bNAbs, “block-and-lock” strategies, CAR T cells, and gene editing, that have shown promise in addressing this challenge. Nevertheless, no researchers have succeeded in fully eradicating or neutralizing the latent reservoir. A notable advantage of CRISPR/Cas9 is its independence from immune system function or antigen expression, meaning that patients can remain on ART during treatment. In contrast, approaches such as “shock-and-kill”, bNAb therapies, and CAR T cells rely on an immune system response to eliminate infected cells (an approach complicated by the immunosuppressive nature of HIV-1 infection). Advancing gene-editing technologies enhance target specificity and minimize unintended genomic alterations, as well as improving the engraftment, survival, and functional persistence of transplanted cells. Innovations in precise gene-editing platforms, such as CRISPR-Cas9 and next-generation genome engineering tools, offer significant potential for generating HIV-resistant cells with reduced risk of off-target consequences. However, several challenges remain, such as mitigating the risk of adaptive immune responses, preventing viral escape mutations, minimizing off-target editing, achieving efficient delivery systems, and ensuring broad applicability across diverse HIV-1 subtypes. Moreover, integrating stem cell-based therapies with adjunctive approaches, such as immunomodulatory treatments or therapeutic vaccines, could amplify the immune response and create synergistic mechanisms for suppressing HIV-1 replication. Also, promising avenues include the development of advanced CAR T-cell therapies with improved specificity for HIV-1 antigens and the application of immune checkpoint inhibitors to bolster T-cell functionality. Breakthroughs in synthetic biology and biomaterials science could further revolutionize the field by enabling the design of synthetic microenvironments that support the survival, proliferation, and long-term activity of transplanted cells in vivo. Such innovations hold the potential to transform stem cell-based HIV therapies into safe, effective, and scalable solutions for achieving sustained viral control or eradication.

Advancements in single-cell technologies have significantly transformed our comprehension of the mechanisms underlying HIV-1 persistence, as well as contributing to the fields of human diseases, immunology, oncology, and development. A key contribution to the HIV-1 field is the application of single-cell multi-omics, which is one of the few methods capable of addressing the heterogeneity and rarity of HIV-1-infected cells, especially those lacking specific markers for enrichment [101]. Genome-wide profiling offers an unbiased means of identifying various cell types and uncovering novel mechanisms. The integration of multi-omic profiling, encompassing epigenetic regulators (DNA via assay for transposase-accessible chromatin with high-throughput sequencing), transcriptional networks (RNA via RNA sequencing), and cellular markers/therapeutic targets (proteins via cellular indexing of transcriptomes and epitopes by sequencing), enhances our understanding of the HIV-1 reservoir through the lens of the central dogma of molecular biology. TCR clonal tracking reveals the clonal dynamics of HIV-1-infected cells. It is essential to exercise caution to ensure that findings are mechanistically meaningful, biologically reproducible, and statistically sound [101]. Bioinformatic analysis of single-cell multi-omics must be tailored to the specific biological questions at hand, rather than relying on default protocols. Accurate identification of rare HIV-1+ cells demands meticulous procedures to prevent sequencing and mapping artifacts [101].

In addition to uses in predictive modeling through machine learning, artificial intelligence has been employed to enhance HIV serodisclosure efforts. In a small pilot study conducted by Muessig et al. developed and assessed the Tough Talks virtual reality program designed to assist young men who have sex with men in role-playing HIV serostatus disclosure, aiming to promote protective behaviors against HIV transmission [102]. The study gathered qualitative data through focus groups with young men who have sex with men living with HIV, exploring their experiences with serodisclosure, which were then used to create a database of common expressions encountered during HIV serostatus discussions. Participants were able to select a virtual character (i.e., an avatar) and engage in various disclosure scenarios using the virtual reality platform. Most participants found the tool acceptable, although some considered it overly complex and cumbersome. A randomized trial is planned to assess the impact of this artificial intelligence tool, comparing its effects when delivered online versus in a clinic setting, on HIV viral load and condomless anal sex behaviors among young men who have sex with men [103]. Chatbots, also known as conversational agents, have been a relatively underexplored application of artificial intelligence in HIV prevention thus far. These chatbots can interact with users anonymously through voice or text messaging, utilizing machine learning algorithms to generate context-appropriate prompts or responses based on prior interactions. The rapid advances in natural language processing, mobile applications, and social media platforms have significantly accelerated the adoption of chatbots [104]. Brixey et al., in 2016, launched a chatbot for sexual health information related to HIV/AIDS (SHIHbot) via Facebook Messenger, offering users information sourced from a database compiled from professional medical and public health materials [105]. The U.S. Department of Health and Human Services piloted, in 2018, a chatbot on Facebook Messenger during the International AIDS Conference, gaining valuable insights into the need for tailored, conversational, and multimedia content for the future development of chatbots aimed at HIV prevention [106]. Beyond delivering HIV-related information, chatbots hold potential for supporting individuals in decisions regarding preexposure prophylaxis use or adherence to preexposure prophylaxis and antiretroviral therapy, though these applications remain unexplored [107].

Ultimately, continual learning from disciplines beyond HIV-1 research, including artificial intelligence, bioinformatics, biotechnology, the human cell atlas, immunology, and cancer biology, will expedite groundbreaking discoveries regarding the mechanisms of HIV-1 persistence and the development of novel therapeutic strategies.

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
