# Peer review of "Breaking Barriers to an HIV-1 Cure: Innovations in Gene Editing, Immune Modulation, and Reservoir Eradication"

_life, 2025, doi:10.3390/life15020276_

Round 1
Reviewer 1 Report
Comments and Suggestions for Authors
This manuscript presents a very well-written summary review of current approaches for generating a “cure” of chronic HIV infection and generally cites up-to-date references. However, the review fails to include the strategy of bispecific antibodies targeting HIV-1-infected cells to promote recruitment and killing of infected cells by CD8+ and NK cells. Mention of fusion peptide-directed monoclonal antibodies particularly VRC34.01, was not included in the discussion of therapeutic neutralizing antibodies. These omissions should be addressed in this otherwise very thorough review of HIV-1 cure strategies. Minor comments that require attention are listed below.
Minor comments:
Line 28: Editing “reducing viral loads to undetectable levels” to “reducing viral loads in plasma to undetectable levels’ is suggested.
Combining paragraphs 1 and 2 under Introduction should be combined. The sentence in line 32 could be edited to “Therefore HIV-1 continues to pose a major global health challenge,” with this first sentence following the last sentence in the first paragraph with the two paragraphs combined.
Lines 104-105: The statement addressing “BRG1/BRM-associated factor (BAF) inhibitors” should include a citation.
Line 298: Statement stating “with donors lacking the” should be corrected to “a donor encoding the”.
Line 327: “immune cells cytotoxic T lymphocytes” should be edited to “immune cytotoxic T lymphocytes”.
Line 435: “In other study it has been employed” should be edited to “Another study employed”.
The section discussing CRISPR-based strategies failed to cite more recent publications including P.K. Dash et al. 120 (19) e2217887120, 2023, PNAS. This more recent report is worthy of mention.
Author Response
Thank you very much for your thoughtful and constructive feedback. I greatly appreciate your positive comments on the manuscript's thoroughness and the up-to-date references cited. I also value your suggestions for improving the manuscript and will address the points you raised.
Regarding the omission of bispecific antibodies targeting HIV-1-infected cells, I agree that this is an important strategy that should be discussed. I will include a section on the role of bispecific antibodies, such as those targeting HIV-1-infected cells to promote the recruitment and killing of infected cells by CD8+ T cells and NK cells (page 15; line 767, references [92-100], in red). Similarly, I will revise the section on therapeutic neutralizing antibodies to include a discussion of fusion peptide-directed monoclonal antibodies, particularly VRC34.01 (page 8; line 404, in red), as you kindly suggested.
For the minor comments (the lines have changed due to the modifications made in the text):
- Line 28: I will edit this sentence as suggested, changing “reducing viral loads to undetectable levels” to “reducing viral loads in plasma to undetectable levels.” (page 1; line 32, in red)
- Introduction: I will combine paragraphs 1 and 2, and the sentence in line 35, (page 1; in red) will be revised to read: "Therefore, HIV-1 continues to pose a major global health challenge."
- Lines 104-105: I will add [14] (page 3; line 118, in red) the appropriate citation for “BRG1/BRM-associated factor (BAF) inhibitors” as per your suggestion.
- Line 298: I will correct the statement to: “a donor encoding the” as you indicated (page 9; line 455, in red).
- Line 327: The phrase “immune cells cytotoxic T lymphocytes” will be edited to “immune cytotoxic T lymphocytes.” (Page 11; line 538, in red).
- Line 435: I will revise this sentence to: “Another study employed” as suggested (page 14; line 679, in red).
- CRISPR-based strategies: I will include the more recent publication by P.K. Dash et al., 2023, PNAS, as you recommended (coments about this publication in page 14; line 706, reference [84], in red).

Reviewer 2 Report
Comments and Suggestions for Authors
The review paper is well written and comprehensive. The paper covers the emerging strategies and each approach has been described with appropriate references. The authors also provide insight into future work that could advance the field.
The paper was easy to follow. A minor formatting comment: Lines 315-325 seem to be in another font and needs to be checked and updated.
The references are appropriate to the paper.
Author Response
Thank you very much for your positive feedback and for taking the time to review the manuscript. I am glad to hear that you found the paper well-written, comprehensive, and easy to follow, and that the emerging strategies and their respective references were appropriately addressed.
Regarding the minor formatting issue you mentioned, I will carefully check and update the font in lines 315-325 (the lines have changed due to the modifications made in the text, then the new page is 11; line 519, in green) to ensure consistency throughout the document.
I appreciate your comment on the references, and I am pleased to know they are appropriate for the paper.
Once again, thank you for your valuable input. I will address the formatting issue promptly, and I am grateful for your thoughtful review.

Reviewer 3 Report
Comments and Suggestions for Authors
The manuscript titled "Breaking Barriers to an HIV-1 Cure: Innovations in Gene Editing, Immune Modulation, and Reservoir Eradication" presents a comprehensive overview of recent advancements in HIV-1 research, focusing on innovative therapeutic strategies. While the manuscript provides valuable insights into innovative strategies for HIV-1 treatment, addressing the following points could significantly enhance its impact and relevance in advancing HIV research and therapy.
:1. The authors mention various therapeutic strategies, such as CRISPR/Cas9 and CAR T-cell therapy; however, they do not provide detailed quantitative data or results from clinical trials.
2: Authors may go into in-depth discussion on the specific mechanisms behind each challenge. For example, the potential off-target effects of gene-editing technologies like CRISPR/Cas9 are mentioned but not elaborated upon, leaving a gap in understanding the implications for clinical application.
3: The authors discuss the immediate therapeutic potentials but do not adequately address the long-term outcomes and sustainability of these treatments. For example, while discussing the "shock and kill" strategy, it notes that HDAC inhibitors have shown limited effects on reducing the latent reservoir without exploring the long-term consequences of such treatments on patient health.
4: Authors may include factors such as genetic differences, co-infections, and pre-existing conditions that can significantly affect treatment outcomes.
5. The explanation of the "shock and kill" approach raises concerns about neurotoxicity caused by immune-mediated clearance in the CNS; however, it does not provide particular strategies for reducing these dangers or alternate approaches to avoid this issue.
6. Although the manuscript mentions cutting-edge methodologies like artificial intelligence and single-cell transcriptomics, it does not explore how these technologies could be integrated into existing strategies or their potential impact on future research directions.
7: In my opinion, it will be better to include figures or diagrams that illustrate complex concepts, such as the mechanisms of action for different therapies or flowcharts showing treatment pathways.
8: Authors should ensure consistent use of terminology throughout the manuscript. For instance, define acronyms at first use and maintain their usage consistently.
Author Response
Thank you for your detailed and thoughtful feedback. I greatly appreciate the time you took to review my manuscript. Your comments are invaluable and will certainly help to enhance the quality and impact of the manuscript. Please find my responses below, addressing each point raised:
1. The authors mention various therapeutic strategies, such as CRISPR/Cas9 and CAR T-cell therapy; however, they do not provide detailed quantitative data or results from clinical trials.
Response to lack of Clinical Data or Results:
I understand the importance of incorporating detailed quantitative data from clinical trials to strengthen the manuscript’s relevance. I will revise the manuscript to include relevant clinical trial data, particularly focusing on the results from studies involving CRISPR/Cas9 and CAR T-cell therapies. This will help provide a clearer picture of the clinical progress and challenges in these innovative therapies. (Page 3; line 119, new reference added [14] in blue; page 4; line 17, new references added [18, 20-22] in blue; page 6; line 262, new references added [31-36] in blue; page 8; line 368, new references added [46, 47] in blue; page 11; line 545, new references added [63]; page 14; line 707, new references added [84] in red.
Text added:
“In an interesanting study shows these BAF complexes facilitate the access of transcription factors and cofactors to enhancers and promoters, thereby regulating gene expression critical for the growth and differentiation of acute myeloid leukemia stem and progenitor cells. In acute myeloid leukemia with MLL1 rearrangement (MLL1r) or mutant NPM1 (mtNPM1), treatment with menin inhibitors (MIs) can induce clinical remissions; however, many patients either exhibit resistance or experience relapse, with some acquiring menin mutations. FHD-286, an orally bioavailable and selective BRG1/BRM inhibitor currently in clinical development for acute myeloid leukemia, has demonstrated potent effects in preclinical studies. FHD-286 induces differentiation and lethality in AML cells with MLL1r or mtNPM1 by disrupting chromatin accessibility and downregulating key factors such as c-Myc, PU.1, and CDK4/6. Combining FHD-286 with decitabine, a bromodomain and extra-terminal domain (BET) inhibitor, a menin inhibitor or venetoclax significantly enhanced acute myeloid leukemia cell lethality in vitro in a synergistic manner. In patient-derived xenograft models of acute myeloid leukemia with MLL1r or mtNPM1, FHD-286 treatment effectively reduced leukemia burden, prolonged survival, and diminished the leukemia-initiating potential of acute myeloid leukemia stem and progenitor cells. Furthermore, co-treatment of FHD-286 with BET inhibitors, menin inhibitors, decitabine, or venetoclax significantly reduced disease burden and improved survival outcomes compared to single-agent therapies, without inducing notable toxicity. These findings underscore the potential of FHD-286-based combination therapies as a promising treatment strategy for acute myeloid leukemia with MLL1r or mtNPM1 [14].”
“Promising preliminary results have been observed with this approach. HDACi have been shown to reverse latency and induce HIV-1 RNA expression in models such as Simian Immunodeficiency Virus (SIV)-infected rhesus macaques [18], in clinical trials [19] and ex vivo patient-derived CD4+ T cells [20]. However, a key limitation of the shock and kill approach is that HDACi have shown limited or no effect on the size of the latent reservoir, both in ex vivo and in vivo studies, as well as in clinical trials [21]. While in vitro studies and animal models suggest that HDACi, such as Suberoylanilide Hydroxamic Acid (SAHA) and panobinostat, can induce HIV transcription and reduce latency, clinical outcomes have been less promising [21]. For instance, despite achieving sustained serum levels with a long-acting injectable system, SAHA did not significantly impact plasma viral RNA or viral suppression. Instead, it increased cell-associated HIV DNA levels, possibly due to enhanced CD4+ T-cell susceptibility and impaired immune responses [21]. Panobinostat showed no measurable effects on the latent reservoir, consistent with its minimal influence on HIV replication in vitro. These findings highlight the need to critically evaluate the long-term consequences of HDACi in HIV-1 patients [22]. Although these agents show theoretical potential, their inability to substantially reduce the reservoir in vivo underscores the complexity of latency and the necessity for alternative therapeutic strategies to effectively target persistent HIV infection.”
“For example, in the J-Lat 9.2 cell model of HIV-1 latency, short hairpin RNAs such as PromA, 143, and PromA/143 demonstrated the ability to maintain viral inactivity even in the presence of LRAs, while keeping the chromatin compacted [31]. This suggests that short hairpin RNAs could confer resistance to provirus reactivation. Similarly, small interfering RNAs si143, which targets tandem NF-κB motifs within the viral 5′LTR, has been shown to induce transcriptional silencing [32]. Small interfering RNAs and short hairpin RNAs hold considerable promise due to their specificity, potency, and adaptability-critical advantages given HIV-1’s high mutational rate [33].
A phase Ib/IIa proof-of-concept trial investigated whether romidepsin, a histone deacetylase inhibitor, could reverse HIV-1 latency in humans while maintaining ART [31]. Six aviremic HIV-1-infected adults received weekly intravenous doses of 5 mg/m² romidepsin for three weeks. The treatment significantly increased histone H3 acetylation (3.7–7.7-fold) and HIV-1 transcription (2.4–5.0-fold, p = 0.03), as measured by unspliced cell-associated HIV-1 RNA. Plasma HIV-1 RNA levels, initially <20 copies/mL, rose to quantifiable levels (46–103 copies/mL, p = 0.04) in five participants after the second infusion. Importantly, romidepsin did not reduce HIV-specific T cell counts or impair their cytokine production. Adverse events were mild (grades 1–2) and aligned with known romidepsin side effects [34, 35]. These findings demonstrate that significant HIV-1 latency reversal is achievable in vivo without compromising T cell-mediated immune responses, marking an important step toward strategies targeting the HIV-1 reservoir.
Another promising molecule in the block and lock strategy is didehydro-cortistatin A, a potent inhibitor of the HIV-1 Tat protein, whose unique structural modifications, enhance its selectivity and potency as a Tat inhibitor. Didehydro-cortistatin A exhibits a high affinity for the transactivation-responsive RNA-binding region of Tat and disrupts Tat-mediated transcriptional feedback loops, effectively "freezing" proviruses into a long-lived, inactive state [34, 35]. Remarkably, didehydro-cortistatin A remains effective even after the cessation of treatment. Its superior pharmacokinetic profile, stability, and ability to cross the blood-brain barrier further distinguish it as a promising candidate for targeting viral reservoirs, particularly in the CNS. These characteristics underscore its unique potential for clinical applications in HIV-1 eradication strategies. Remarkably, didehydro-cortistatin A remains effective even after the cessation of treatment [35]. In vivo studies in the bone marrow-liver-thymus mouse model demonstrated that didehydro-cortistatin A suppressed HIV expression by inducing epigenetic silencing via restricted recruitment of RNA polymerase II to the promoter [36]. Didehydro-cortistatin A also alters Tat's protein environment, enhancing its resistance to proteolytic degradation and interfering with the Tat-TAR interaction, which is essential for Tat functionality. Consequently, didehydro-cortistatin A prevents proviral reactivation [36].”
“Other studies in Phase 1 have also evaluated additional bNAbs such as PGT121, which targets the V3 loop base, and VRC07-523LS, with results from these studies expected shortly [46].
Combination therapies using bNAbs targeting non-overlapping epitopes, such as 3BNC117 and 10–1074, have shown promise in extending antiviral coverage. Among viremic participants receiving one or three infusions of this combination, those with antibody-sensitive viruses experienced greater declines in viremia compared to monotherapy, with reductions sustained for up to three months. Complete suppression was observed only in participants with low baseline viral loads, and resistance to 3BNC117 was not observed despite persistent viremia and viral recombination. However, the shorter half-life of 3BNC117 resulted in a transition to 10–1074 monotherapy, coinciding with the emergence of 10–1074-resistant variants [47].
In ART-suppressed individuals harboring bNAb-sensitive viruses, the combination of 3BNC117 and 10–1074 effectively maintained suppression during ATI. The median rebound time for seven out of nine participants was 21 weeks (range: 15–26 weeks), with two participants maintaining suppression beyond 30 weeks [47].”
“In these trials, CCR5 is a critical coreceptor for HIV entry. This study, for example, assessed the safety of gene editing using zinc-finger nucleases to permanently disrupt the CCR5 gene in autologous CD4 T cells, followed by their infusion into patients with chronic aviremic HIV. In this case, twelve patients received a single infusion of 10 billion autologous CD4 T cells, 11–28% of which were genetically modified with zinc-finger nucleases. Six patients underwent a treatment interruption of ART four weeks post-infusion. Safety, immune reconstitution, and HIV resistance were evaluated. As results, there was one serious adverse event that occurred, attributed to a transfusion reaction. Median CD4 T-cell counts rose significantly from 448 to 1517 cells/mm³ at week 1 (P<0.001). CCR5-modified cells represented 8.8% of peripheral mononuclear cells and 13.9% of CD4 T cells, with an estimated half-life of 48 weeks. During ART interruption, CCR5-modified cells declined more slowly than unmodified cells (-1.81 vs. -7.25 cells/day, P=0.02). HIV RNA became undetectable in one of four evaluable patients, and most showed reduced HIV DNA levels. Infusions of CCR5-modified CD4 T cells appear safe and show potential for immune reconstitution and HIV control [63].”
- Authors may go into in-depth discussion on the specific mechanisms behind each challenge. For example, the potential off-target effects of gene-editing technologies like CRISPR/Cas9 are mentioned but not elaborated upon, leaving a gap in understanding the implications for clinical application.
Response to in-depth Discussion on Specific Mechanism:
I appreciate your suggestion to elaborate on the potential off-target effects of CRISPR/Cas9 and other gene-editing technologies. I will expand this section to provide a more detailed discussion of these risks, the strategies being developed to minimize off-target effects, and their implications for clinical applications, ensuring a better understanding of the safety profiles for these technologies (page 10; line 460, new references added [54-57], in purple; page 12; line 581, new references added [63, 64], in blue; page 15; line 724, new references added [85-91], in blue).
Text added:
“While promising, this strategy is associated with significant risks and limitations. In the context of medical conditions, the primary challenges include disease relapse, infectious complications, and regimen-related toxicities, particularly among older patient populations [54]. Careful consideration must be given to quality of life and long-term outcomes, especially in elderly individuals. Access to HSCT is influenced by socioeconomic status, education level, and ethnicity, with migrants and minority groups frequently encountering systemic barriers [54]. Additionally, cultural beliefs and language barriers may hinder patient comprehension and adherence to HSCT protocols. Immigrant populations often exhibit distinct perceptions of illness and treatment, which can impact their interaction with and utilization of healthcare systems [54].
HSCT raises significant ethical concerns, particularly in the domains of informed consent, donor-recipient matching, and the long-term implications of transplantation. The decision-making process for HSCT is inherently complex, involving multiple stakeholders, including patients, families, and healthcare providers [55]. This process necessitates a careful balance between potential risks, clinical benefits, and the overall impact on the patient's quality of life. Pediatric HSCT introduces additional ethical challenges, particularly concerning the involvement of minors in decision-making processes and the long-term physical, psychological, and social effects on survivors. A particularly contentious issue is the right of mature minors to refuse life-sustaining treatments, such as HSCT, which often intersects with legal, ethical, and familial considerations. Furthermore, conflicts of interest in HSCT research and clinical practice may arise from various sources, including economic pressures, selective publication of data, and the prioritization of research that is financially lucrative over studies addressing unmet medical needs [55]. These conflicts can undermine the integrity of research and the equitable delivery of care. Ethical oversight committees play a pivotal role in addressing these challenges by safeguarding the rights and safety of research participants. These committees are responsible for ensuring compliance with internationally recognized ethical guidelines, such as the Declaration of Helsinki [55]. To maintain their credibility and effectiveness, ethical committees must operate independently and ensure that the benefits and risks of research are equitably distributed across all societal groups, thereby promoting justice and inclusivity in scientific advancement.
HSCT is a highly invasive and morbid procedure, often associated with graft-versus-host disease and challenges in finding HLA-compatible CCR5Δ32/Δ32 donors [46]. Furthermore, the strategy does not protect against HIV variants utilizing the CXCR4 co-receptor for cell entry, leaving the potential for continued viral replication [56]. These challenges underscore the need for careful consideration and further research to improve the safety and feasibility of this approach.
Patients infected with CXCR4-tropic HIV generally exhibit a poorer clinical prognosis and would not benefit from the transplantation of CCR5Δ32 hematopoietic stem and progenitor cells, as the CXCR4-tropic virus can independently infect cells without relying on the CCR5 co-receptor [57]. There is a critical need for strategies to combat the multiple variants of HIV that evolve in each patient, as well as the identification of therapies effective against both CCR5- and CXCR4-tropic HIV-1. Developing diverse genetic resistance mechanisms is comparable to the requirement for maintaining multiple small-molecule inhibitors during ART to control viral replication [57]. The development of an autologous hematopoietic stem cell therapy, which could improve transplantation safety, enhance treatment efficacy by providing resistance to both dual- and CXCR4-tropic HIV, and expand the pool of HIV patients eligible for HSCT.”
“Issues with CAR T-cell persistence and lack of robust expansion have also been reported [63]. Another critical challenge is the risk of off-target effects, as evidenced by B-cell aplasia observed in leukemia studies targeting the B-cell antigen CD19 with CAR T cells [64].
Addressing these limitations is essential to realize the full therapeutic potential of CAR T cells in HIV-1 treatment. If these obstacles can be overcome, CAR T-cell therapies hold significant promise as a transformative approach to combating HIV-1. Ongoing research focuses on optimizing processes for immune cell selection and expansion, leveraging cytokines and growth factors to enhance cellular proliferation and functional capacity [58]. Another major hurdle is ensuring the in vivo persistence and proper trafficking of infused cells. For therapeutic efficacy, these cells must survive and localize to sites of HIV replication, such as lymphoid tissues and viral reservoirs. Strategies to address this include the genetic modification of T cells to express anti-apoptotic genes and the use of adjunctive therapies to improve cell survival and function [58]. Additionally, advances in imaging technologies are providing critical insights into the distribution and longevity of infused cells, aiding in the refinement of therapeutic protocols. In conclusion, cell-based immunotherapies represent a transformative approach to HIV-1 treatment, with the potential to enhance antiviral immunity and selectively target HIV-infected cells. While substantial challenges remain, ongoing research and technological progress are steadily addressing these obstacles. Continued exploration of novel strategies, combined with rigorous clinical evaluation, will be essential for fully realizing the therapeutic potential of these approaches in combating HIV-1.”
“These off-target genome editing refers to unintended genetic modifications caused by engineered nucleases, which can still occur in DNA sequences with minor mismatches in the Protospacer Adjacent Motif (PAM)-distal region of the sgRNA [84]. These off-target sites contain noncanonical PAMs and diverse nucleotide variations, with mismatches being more tolerated at the 5′ end than the 3′ end of the gRNA [85]. A mismatch in the seed region can hinder Cas9 activation, while three or more mismatches can disrupt HNH conformation, preventing cleavage. The structural properties of gRNAs significantly influence off-target effects, which not only reduce CRISPR-Cas9's therapeutic precision but also compromise gene function studies [86]. These effects may lead to adverse consequences such as unintended DNA damage, immune activation, and cytotoxicity.
Recent strategies to mitigate CRISPR-Cas9's off-target effects focus on optimizing sgRNA design, engineering Cas variants, and employing novel editing systems and inhibitors:
-Improving sgRNA Specificity: adjustments to sgRNA, such as optimizing GC content (40-60%), shortening sgRNA sequences (<20 nucleotides), or chemically modifying the sgRNA backbone, have enhanced target specificity [85]. Techniques like the "GG20" method, replacing bases at the sgRNA 5′ end with guanines, further minimize off-target interactions without sacrificing on-target efficiency [87].
-Enhanced Cas Variants: engineered Cas9 mutants like enhanced specificity Cas9 (eSpCas9) and SpCas9-HF1 exhibit higher fidelity by reducing non-specific DNA interactions while retaining on-target activity. Cas9 nickase variants, which cleave single DNA strands, also reduce collateral damage and improve accuracy. Additionally, Cas9 homologs requiring rare PAM sequences, such as SaCas9, offer precise targeting with reduced off-target potential [88].
-Prime Editing: prime editing eliminates the need for double-strand breaks and donor DNA, significantly lowering off-target risks. Utilizing nicking Cas9 (nCas9), reverse transcriptase, and pegRNA, this system achieves precise base editing with minimal unintended consequences, though its efficiency remains a challenge [89].
-Anti-CRISPR Proteins: Anti-CRISPR proteins inhibit CRISPR-Cas activity by blocking DNA binding or cleavage, enhancing target selectivity, and mitigating off-target effects. Anti-CRISPR proteins provide a safety mechanism for genome editing but require further exploration for broader applications [90].
-SuperFi-Cas9: the recently developed (Super Fidelity Cas9) SuperFi-Cas9, engineered to disrupt mismatch stabilization, exhibits 4000-fold improved fidelity compared to wild-type Cas9. Early studies demonstrate its high specificity, though its in vivo applications remain limited [91].
These advancements collectively represent significant progress in reducing off-target genome editing effects, paving the way for safer and more precise CRISPR-based therapies”.
- The authors discuss the immediate therapeutic potentials but do not adequately address the long-term outcomes and sustainability of these treatments. For example, while discussing the "shock and kill" strategy, it notes that HDAC inhibitors have shown limited effects on reducing the latent reservoir without exploring the long-term consequences of such treatments on patient health.
Response to long-term Outcomes and Sustainability:
I acknowledge the importance of addressing long-term outcomes and the sustainability of this therapy. I will revise the manuscript to include a more thorough examination of the long-term effects in "shock and kill" strategy, with a focus on both the potential benefits and drawbacks. This will include a discussion on patient health and side effects (page 4; line 174, new refernces added [21, 22] in blue)
Text added:
“a key limitation of the shock and kill approach is that HDACi have shown limited or no effect on the size of the latent reservoir, both in ex vivo and in vivo studies, as well as in clinical trials [21]. While in vitro studies and animal models suggest that HDACi, such as Suberoylanilide Hydroxamic Acid (SAHA) and panobinostat, can induce HIV transcription and reduce latency, clinical outcomes have been less promising [21]. For instance, despite achieving sustained serum levels with a long-acting injectable system, SAHA did not significantly impact plasma viral RNA or viral suppression. Instead, it increased cell-associated HIV DNA levels, possibly due to enhanced CD4+ T-cell susceptibility and impaired immune responses [21]. Panobinostat showed no measurable effects on the latent reservoir, consistent with its minimal influence on HIV replication in vitro. These findings highlight the need to critically evaluate the long-term consequences of HDACi in HIV-1 patients [22]. Although these agents show theoretical potential, their inability to substantially reduce the reservoir in vivo underscores the complexity of latency and the necessity for alternative therapeutic strategies to effectively target persistent HIV infection.”
- Authors may include factors such as genetic differences, co-infections, and pre-existing conditions that can significantly affect treatment outcomes.
Thank you for your insightful suggestion regarding the inclusion of factors such as genetic differences, co-infections, and pre-existing conditions that may influence treatment outcomes. I fully acknowledge the significance of these factors in shaping therapeutic responses and agree that they represent crucial aspects to consider in the context of HIV-1 cure strategies. However, a comprehensive discussion on these variables would require an in-depth exploration that extends beyond the current scope of this manuscript, as it would necessitate a separate review focusing specifically on the interplay between host factors and therapeutic efficacy. Given the extensive nature of this topic, I plan to address it thoroughly in a future dedicated study. I sincerely appreciate your valuable input, which has provided me with important considerations for my ongoing and future research efforts.
- The explanation of the "shock and kill" approach raises concerns about neurotoxicity caused by immune-mediated clearance in the CNS; however, it does not provide particular strategies for reducing these dangers or alternate approaches to avoid this issue. Response to neurotoxicity in the Shock and Kill Approach:
I recognize the concern regarding neurotoxicity associated with immune-mediated clearance in the CNS. I will revise the manuscript to propose potential strategies for mitigating this risk, such as targeted delivery methods or adjunct therapies that could reduce neurotoxicity. Additionally, I will explore alternative approaches that could circumvent this issue (page 5; line 214, new reference added [27], in blue).
Text added:
“Longitudinal assessments rely on viral RNA levels and biomarkers of neuroinflammation and neuronal injury in cerebrospinal fluid. Although this approach is valuable, it is limited due to the invasive nature of lumbar punctures [27]. Less invasive alternatives, such as imaging techniques, have emerged as potential tools for studying CNS effects in vivo. Magnetic resonance imaging, while widely used, does not provide cellular-level resolution or detect changes such as neuronal death. More informative techniques include nuclear imaging approaches, such as single-photon emission computed tomography and positron emission tomography. These modalities use radioactive tracers to detect immune activation, inflammation, and neuronal injury associated with HIV infection. For instance, positron emission tomography imaging of macrophage colony‐stimulating factor 1 receptor has been shown to track microglial neuroinflammation, and tracers for monitoring synaptic density are available [27]. However, to directly visualize HIV-infected cells in the CNS, the development of HIV-specific tracers capable of penetrating the blood-brain barrier is urgently needed. Another promising technique is metabolic imaging via magnetic resonance spectroscopy, which measures chemical changes in neurometabolites, allowing the monitoring of neuroinflammation and associated neuronal injury. However, magnetic resonance spectroscopy is limited to a small number of brain regions, and its utility can be confounded by comorbidities. Overall, imaging techniques hold great promise for evaluating the effects and safety of HIV eradication strategies on the CNS [27]. Despite their limitations in directly monitoring HIV reactivation, the continued development and application of these advanced imaging modalities remain critical for advancing our understanding of CNS-related HIV pathology and the impact of eradication strategies”.
- Although the manuscript mentions cutting-edge methodologies like artificial intelligence and single-cell transcriptomics, it does not explore how these technologies could be integrated into existing strategies or their potential impact on future research directions.
Response to integration of Cutting-edge Technologies:
I appreciate your suggestion to delve deeper into the integration of artificial intelligence and single-cell transcriptomics into current HIV-1 cure strategies. I will expand this section to explore how these technologies can be used to optimize therapeutic strategies, such as identifying novel targets, enhancing precision in gene-editing techniques, or improving patient monitoring in clinical trials (page 18; line 860, new references added [101-107], in blue).
Text added:
“Advancements in single-cell technologies have significantly transformed our comprehension of the mechanisms underlying HIV-1 persistence, as well as contributing to the fields of human diseases, immunology, oncology, and development. A key contribution to the HIV-1 field is the application of single-cell multi-omics, which is one of the few methods capable of addressing the heterogeneity and rarity of HIV-1-infected cells, especially those lacking specific markers for enrichment [101]. Genome-wide profiling offers an unbiased means to identify various cell types and uncover novel mechanisms. The integration of multi-omic profiling, encompassing epigenetic regulators (DNA via Assay for Transposase-Accessible Chromatin with high-throughput sequencing), transcriptional networks (RNA via RNA-sequencing), and cellular markers/therapeutic targets (proteins via Cellular Indexing of Transcriptomes and Epitopes by sequencing), enhances our understanding of the HIV-1 reservoir through the lens of the central dogma of molecular biology. TCR clonal tracking reveals the clonal dynamics of HIV-1-infected cells. It is essential to exercise caution to ensure that findings are mechanistically meaningful, biologically reproducible, and statistically sound [101]. Bioinformatic analysis of single-cell multi-omics must be tailored to the specific biological questions at hand, rather than relying on default protocols. Accurate identification of rare HIV-1+ cells demands meticulous procedures to prevent sequencing and mapping artifacts [101].
In addition to the use in predictive modeling through machine learning, artificial intelligence has been employed to enhance HIV serodisclosure efforts. In a small pilot study conducted by Muessig et al., developed and assessed the Tough Talks virtual reality program designed to assist young men who have sex with men in role-playing HIV serostatus disclosure, aiming to promote protective behaviors against HIV transmission [102]. The study gathered qualitative data through focus groups with young men who have sex with men living with HIV, exploring their experiences with serodisclosure, which were then used to create a database of common expressions encountered during HIV serostatus discussions. Participants were able to select a virtual character (i.e., an avatar) and engage in various disclosure scenarios using the virtual reality platform. Most participants found the tool acceptable, although some considered it overly complex and cumbersome. A randomized trial is planned to assess the impact of this artificial intelligence tool, comparing its effects when delivered online versus in a clinic setting, on HIV viral load and condomless anal sex behaviors among young men who have sex with men [103]. Chatbots, also known as conversational agents, have been a relatively underexplored application of artificial intelligence in HIV prevention thus far. These chatbots can interact with users anonymously through voice or text messaging, utilizing machine learning algorithms to generate context-appropriate prompts or responses based on prior interactions. The rapid advances in natural language processing, mobile applications and social media platforms have significantly accelerated the adoption of chatbots [104]. Brixey et al., in 2016, launched a chatbot for sexual health information related to HIV/AIDS (SHIHbot) via Facebook Messenger, offering users information sourced from a database compiled from professional medical and public health materials [105]. The U.S. Department of Health and Human Services piloted, in 2018, a chatbot on Facebook Messenger during the International AIDS Conference, gaining valuable insights into the need for tailored, conversational, and multimedia content for the future development of chatbots aimed at HIV prevention [106]. Beyond delivering HIV-related information, chatbots hold potential for supporting individuals in decisions regarding preexposure prophylaxis use or adherence to preexposure prophylaxis and antiretroviral therapy, though these applications remain unexplored [107].
Ultimately, continual learning from disciplines beyond HIV-1 research, including artificial intelligence, bioinformatics, biotechnology, the human cell atlas, immunology, and cancer biology, will expedite groundbreaking discoveries regarding the mechanisms of HIV-1 persistence and the development of novel therapeutic strategies”.
- Although the manuscript mentions cutting-edge methodologies like artificial intelligence and single-cell transcriptomics, it does not explore how these technologies could be integrated into existing strategies or their potential impact on future research directions.
Response to Figures:
I agree that visual aids would significantly enhance the manuscript, especially for complex concepts. I will include figures that illustrate the mechanisms of action for different therapies and provide flowcharts showing treatment pathways, which will help clarify the information for readers. (Figure added at the end of manuscript).
- Authors should ensure consistent use of terminology throughout the manuscript. For instance, define acronyms at first use and maintain their usage consistently. Response to consistency of Terminology:
I will ensure consistent use of terminology throughout the manuscript, including defining acronyms at the end of the paper and maintaining their consistency in subsequent mentions. This will improve clarity and readability.
Once again, I would like to express my gratitude for your insightful comments and recommendations. I believe these revisions will substantially improve the manuscript and its contribution to advancing the field of HIV-1 research and therapy.

Reviewer 4 Report
Comments and Suggestions for Authors
Authors can improved based on the comments given to publish the review article in better shape.

Author Response
I sincerely appreciate your thoughtful and constructive feedback on my manuscript. Your insightful comments have been invaluable in refining my discussion and enhancing the clarity and depth of the review.
I have carefully considered each of your suggestions and have revised the manuscript accordingly The lines have changed due to the modifications made in the text:
- Line 46: I have expanded our discussion on state-of-the-art methodologies, including single-cell transcriptomics and artificial intelligence, to provide readers with a clearer understanding of their applications in HIV-1 research (page 18; line 861, new references added [101-107], in blue).
Text added:
“Advancements in single-cell technologies have significantly transformed our comprehension of the mechanisms underlying HIV-1 persistence, as well as contributing to the fields of human diseases, immunology, oncology, and development. A key contribution to the HIV-1 field is the application of single-cell multi-omics, which is one of the few methods capable of addressing the heterogeneity and rarity of HIV-1-infected cells, especially those lacking specific markers for enrichment [101]. Genome-wide profiling offers an unbiased means to identify various cell types and uncover novel mechanisms. The integration of multi-omic profiling, encompassing epigenetic regulators (DNA via Assay for Transposase-Accessible Chromatin with high-throughput sequencing), transcriptional networks (RNA via RNA-sequencing), and cellular markers/therapeutic targets (proteins via Cellular Indexing of Transcriptomes and Epitopes by sequencing), enhances our understanding of the HIV-1 reservoir through the lens of the central dogma of molecular biology. TCR clonal tracking reveals the clonal dynamics of HIV-1-infected cells. It is essential to exercise caution to ensure that findings are mechanistically meaningful, biologically reproducible, and statistically sound [101]. Bioinformatic analysis of single-cell multi-omics must be tailored to the specific biological questions at hand, rather than relying on default protocols. Accurate identification of rare HIV-1+ cells demands meticulous procedures to prevent sequencing and mapping artifacts [101].
In addition to the use in predictive modeling through machine learning, artificial intelligence has been employed to enhance HIV serodisclosure efforts. In a small pilot study conducted by Muessig et al., developed and assessed the Tough Talks virtual reality program designed to assist young men who have sex with men in role-playing HIV serostatus disclosure, aiming to promote protective behaviors against HIV transmission [102]. The study gathered qualitative data through focus groups with young men who have sex with men living with HIV, exploring their experiences with serodisclosure, which were then used to create a database of common expressions encountered during HIV serostatus discussions. Participants were able to select a virtual character (i.e., an avatar) and engage in various disclosure scenarios using the virtual reality platform. Most participants found the tool acceptable, although some considered it overly complex and cumbersome. A randomized trial is planned to assess the impact of this artificial intelligence tool, comparing its effects when delivered online versus in a clinic setting, on HIV viral load and condomless anal sex behaviors among young men who have sex with men [103]. Chatbots, also known as conversational agents, have been a relatively underexplored application of artificial intelligence in HIV prevention thus far. These chatbots can interact with users anonymously through voice or text messaging, utilizing machine learning algorithms to generate context-appropriate prompts or responses based on prior interactions. The rapid advances in natural language processing, mobile applications and social media platforms have significantly accelerated the adoption of chatbots [104]. Brixey et al., in 2016, launched a chatbot for sexual health information related to HIV/AIDS (SHIHbot) via Facebook Messenger, offering users information sourced from a database compiled from professional medical and public health materials [105]. The U.S. Department of Health and Human Services piloted, in 2018, a chatbot on Facebook Messenger during the International AIDS Conference, gaining valuable insights into the need for tailored, conversational, and multimedia content for the future development of chatbots aimed at HIV prevention [106]. Beyond delivering HIV-related information, chatbots hold potential for supporting individuals in decisions regarding preexposure prophylaxis use or adherence to preexposure prophylaxis and antiretroviral therapy, though these applications remain unexplored [107].
Ultimately, continual learning from disciplines beyond HIV-1 research, including artificial intelligence, bioinformatics, biotechnology, the human cell atlas, immunology, and cancer biology, will expedite groundbreaking discoveries regarding the mechanisms of HIV-1 persistence and the development of novel therapeutic strategies.”
- Line 118: I have provided an updated description of latency-reversing agent (LRA) classes, detailing their unique contributions and mechanisms of action to strengthen this section (page 2; line 43, in purple)
Text added:
“Latent reservoirs refer to a population of long-lived cells, primarily resting memory CD4⁺ T cells, that harbor integrated but transcriptionally silent HIV provirus. These reservoirs persist despite ART because the virus remains in a dormant state, evading immune detection and viral replication. Upon cellular activation, the latent virus can reactivate, leading to viral rebound if ART is interrupted. Latent reservoirs represent a major barrier to HIV cure efforts.”
- Line 121: The transition between the LRA mechanism and the shock and kill strategy has been refined to better illustrate the interplay between these approaches and their therapeutic utility.
- Line 132: I have included a discussion on potential research directions to test these strategies in in vivo models, outlining key considerations for their preclinical and clinical evaluation (page 4; line 171, new reference added [18, 20-22], in blue).
Text added:
“Promising preliminary results have been observed with this approach. HDACi have been shown to reverse latency and induce HIV-1 RNA expression in models such as Simian Immunodeficiency Virus (SIV)-infected rhesus macaques [18], in clinical trials [19] and ex vivo patient-derived CD4+ T cells [20]. However, a key limitation of the shock and kill approach is that HDACi have shown limited or no effect on the size of the latent reservoir, both in ex vivo and in vivo studies, as well as in clinical trials [21]. While in vitro studies and animal models suggest that HDACi, such as Suberoylanilide Hydroxamic Acid (SAHA) and panobinostat, can induce HIV transcription and reduce latency, clinical outcomes have been less promising [21]. For instance, despite achieving sustained serum levels with a long-acting injectable system, SAHA did not significantly impact plasma viral RNA or viral suppression. Instead, it increased cell-associated HIV DNA levels, possibly due to enhanced CD4+ T-cell susceptibility and impaired immune responses [21]. Panobinostat showed no measurable effects on the latent reservoir, consistent with its minimal influence on HIV replication in vitro. These findings highlight the need to critically evaluate the long-term consequences of HDACi in HIV-1 patients [22]. Although these agents show theoretical potential, their inability to substantially reduce the reservoir in vivo underscores the complexity of latency and the necessity for alternative therapeutic strategies to effectively target persistent HIV infection.”
- Line 149: I have elaborated on the risks of neurotoxicity associated with immune-mediated clearance in the central nervous system and discussed possible mitigation strategies for future research (page 5; line 214, new reference added [27], in blue).
Text added:
“Longitudinal assessments rely on viral RNA levels and biomarkers of neuroinflammation and neuronal injury in cerebrospinal fluid. Although this approach is valuable, it is limited due to the invasive nature of lumbar punctures [27]. Less invasive alternatives, such as imaging techniques, have emerged as potential tools for studying CNS effects in vivo. Magnetic resonance imaging, while widely used, does not provide cellular-level resolution or detect changes such as neuronal death. More informative techniques include nuclear imaging approaches, such as single-photon emission computed tomography and positron emission tomography. These modalities use radioactive tracers to detect immune activation, inflammation, and neuronal injury associated with HIV infection. For instance, positron emission tomography imaging of macrophage colony‐stimulating factor 1 receptor has been shown to track microglial neuroinflammation, and tracers for monitoring synaptic density are available [27]. However, to directly visualize HIV-infected cells in the CNS, the development of HIV-specific tracers capable of penetrating the blood-brain barrier is urgently needed. Another promising technique is metabolic imaging via magnetic resonance spectroscopy, which measures chemical changes in neurometabolites, allowing the monitoring of neuroinflammation and associated neuronal injury. However, magnetic resonance spectroscopy is limited to a small number of brain regions, and its utility can be confounded by comorbidities. Overall, imaging techniques hold great promise for evaluating the effects and safety of HIV eradication strategies on the CNS [27]. Despite their limitations in directly monitoring HIV reactivation, the continued development and application of these advanced imaging modalities remain critical for advancing our understanding of CNS-related HIV pathology and the impact of eradication strategies.”
- Line 174: I have clarified the unique characteristics of didehydro-cortistatin A compared to other molecules, emphasizing its distinct potential for clinical applications. Additionally, I have incorporated quantitative data on the efficacy of LEDGINs molecules (page. ; line 261, new references added [31-37], in blue-first paragraph and in purple-second paragraph)
Text added:
“For example, in the J-Lat 9.2 cell model of HIV-1 latency, short hairpin RNAs such as PromA, 143, and PromA/143 demonstrated the ability to maintain viral inactivity even in the presence of LRAs, while keeping the chromatin compacted [31]. This suggests that short hairpin RNAs could confer resistance to provirus reactivation. Similarly, small interfering RNAs si143, which targets tandem NF-κB motifs within the viral 5′LTR, has been shown to induce transcriptional silencing [32]. Small interfering RNAs and short hairpin RNAs hold considerable promise due to their specificity, potency, and adaptability-critical advantages given HIV-1’s high mutational rate [33].
A phase Ib/IIa proof-of-concept trial investigated whether romidepsin, a histone deacetylase inhibitor, could reverse HIV-1 latency in humans while maintaining ART [31]. Six aviremic HIV-1-infected adults received weekly intravenous doses of 5 mg/m² romidepsin for three weeks. The treatment significantly increased histone H3 acetylation (3.7–7.7-fold) and HIV-1 transcription (2.4–5.0-fold, p = 0.03), as measured by unspliced cell-associated HIV-1 RNA. Plasma HIV-1 RNA levels, initially <20 copies/mL, rose to quantifiable levels (46–103 copies/mL, p = 0.04) in five participants after the second infusion. Importantly, romidepsin did not reduce HIV-specific T cell counts or impair their cytokine production. Adverse events were mild (grades 1–2) and aligned with known romidepsin side effects [34, 35]. These findings demonstrate that significant HIV-1 latency reversal is achievable in vivo without compromising T cell-mediated immune responses, marking an important step toward strategies targeting the HIV-1 reservoir.
Another promising molecule in the block and lock strategy is didehydro-cortistatin A, a potent inhibitor of the HIV-1 Tat protein, whose unique structural modifications, enhance its selectivity and potency as a Tat inhibitor. Didehydro-cortistatin A exhibits a high affinity for the transactivation-responsive RNA-binding region of Tat and disrupts Tat-mediated transcriptional feedback loops, effectively "freezing" proviruses into a long-lived, inactive state [34, 35]. Remarkably, didehydro-cortistatin A remains effective even after the cessation of treatment. Its superior pharmacokinetic profile, stability, and ability to cross the blood-brain barrier further distinguish it as a promising candidate for targeting viral reservoirs, particularly in the CNS. These characteristics underscore its unique potential for clinical applications in HIV-1 eradication strategies. Remarkably, didehydro-cortistatin A remains effective even after the cessation of treatment [35]. In vivo studies in the bone marrow-liver-thymus mouse model demonstrated that didehydro-cortistatin A suppressed HIV expression by inducing epigenetic silencing via restricted recruitment of RNA polymerase II to the promoter [36]. Didehydro-cortistatin A also alters Tat's protein environment, enhancing its resistance to proteolytic degradation and interfering with the Tat-TAR interaction, which is essential for Tat functionality. Consequently, didehydro-cortistatin A prevents proviral reactivation [36].
LEDGINs are other promising class of small molecules that target the LEDGF/p75 binding pocket on HIV-1 integrase. LEDGINs inhibit the interaction between LEDGF/p75 and integrase, which has been shown in vitro to increase the fraction of integrated provirus with a transcriptionally silent phenotype. For LEDGINs, their efficacy is contingent on early administration, as their therapeutic window is limited [37]. A study employed a single-cell branched DNA imaging technique to simultaneously detect viral DNA and RNA, allowing for a detailed assessment of the impact of LEDGIN treatment on HIV-1 integration, transcription, and reactivation in both cell lines and primary cells. These findings demonstrated that LEDGIN-mediated retargeting reduces basal transcriptional activity and impairs proviral reactivation, as evidenced by a significant decrease in viral RNA expression per residual provirus. The interaction between HIV-1 integrase and the chromatin tethering factor LEDGF/p75 serves as a key determinant of integration site preference [37]. By employing LEDGINs to disrupt this interaction, researchers have elucidated how integration retargeting influences the three-dimensional positioning of the provirus, its transcriptional activity, and its susceptibility to reactivation [37]. These results support the feasibility of a "block-and-lock" strategy aimed at permanently silencing HIV-1 by directing integration into genomic regions refractory to reactivation following ART discontinuation.”
- Line 238: I have expanded on the rationale behind targeting non-overlapping epitopes, highlighting how this approach enhances efficacy and minimizes resistance development (page 8; line 392, new reference added [48], in purple).
Text added:
“Targeting non-overlapping epitopes in HIV-1 therapy enhances efficacy and reduces resistance by preventing viral escape through multiple simultaneous mutations, which is often detrimental to viral fitness and replication. This makes it significantly harder for HIV-1 to escape immune surveillance [48]. This approach engages different immune mechanisms, including bNAbs and cytotoxic T cells, leading to synergistic neutralization and improved viral clearance. This multi-pronged attack increases therapeutic potency. Additionally, it provides broader coverage against diverse HIV-1 strains, reducing the risk of treatment failure due to strain variability, and ensures a more durable antiviral response. By making it harder for the virus to evade immune detection, this strategy strengthens long-term viral suppression and supports functional cure efforts [48].”
- Line 302: Ethical concerns and resource limitations in identifying suitable donors have been addressed, along with a discussion on alternative gene-editing approaches to circumvent these challenges. (Page 10; line 460, new references added [54-56], in purple).
Text added:
“While promising, this strategy is associated with significant risks and limitations. In the context of medical conditions, the primary challenges include disease relapse, infectious complications, and regimen-related toxicities, particularly among older patient populations [54]. Careful consideration must be given to quality of life and long-term outcomes, especially in elderly individuals. Access to HSCT is influenced by socioeconomic status, education level, and ethnicity, with migrants and minority groups frequently encountering systemic barriers [54]. Additionally, cultural beliefs and language barriers may hinder patient comprehension and adherence to HSCT protocols. Immigrant populations often exhibit distinct perceptions of illness and treatment, which can impact their interaction with and utilization of healthcare systems [54].
HSCT raises significant ethical concerns, particularly in the domains of informed consent, donor-recipient matching, and the long-term implications of transplantation. The decision-making process for HSCT is inherently complex, involving multiple stakeholders, including patients, families, and healthcare providers [55]. This process necessitates a careful balance between potential risks, clinical benefits, and the overall impact on the patient's quality of life. Pediatric HSCT introduces additional ethical challenges, particularly concerning the involvement of minors in decision-making processes and the long-term physical, psychological, and social effects on survivors. A particularly contentious issue is the right of mature minors to refuse life-sustaining treatments, such as HSCT, which often intersects with legal, ethical, and familial considerations. Furthermore, conflicts of interest in HSCT research and clinical practice may arise from various sources, including economic pressures, selective publication of data, and the prioritization of research that is financially lucrative over studies addressing unmet medical needs [55]. These conflicts can undermine the integrity of research and the equitable delivery of care. Ethical oversight committees play a pivotal role in addressing these challenges by safeguarding the rights and safety of research participants. These committees are responsible for ensuring compliance with internationally recognized ethical guidelines, such as the Declaration of Helsinki [55]. To maintain their credibility and effectiveness, ethical committees must operate independently and ensure that the benefits and risks of research are equitably distributed across all societal groups, thereby promoting justice and inclusivity in scientific advancement.
HSCT is a highly invasive and morbid procedure, often associated with graft-versus-host disease and challenges in finding HLA-compatible CCR5Δ32/Δ32 donors [46]. Furthermore, the strategy does not protect against HIV variants utilizing the CXCR4 co-receptor for cell entry, leaving the potential for continued viral replication [56]. These challenges underscore the need for careful consideration and further research to improve the safety and feasibility of this approach.
- Line 304: I have included a discussion on the impact of CXCR4-tropic HIV on the broader applicability of gene therapy strategies, emphasizing the need for screening and potential additional interventions (page 10; line 498, new reference added [57], in purple).
Text added:
“Patients infected with CXCR4-tropic HIV generally exhibit a poorer clinical prognosis and would not benefit from the transplantation of CCR5Δ32 hematopoietic stem and progenitor cells, as the CXCR4-tropic virus can independently infect cells without relying on the CCR5 co-receptor [57]. There is a critical need for strategies to combat the multiple variants of HIV that evolve in each patient, as well as the identification of therapies effective against both CCR5- and CXCR4-tropic HIV-1. Developing diverse genetic resistance mechanisms is comparable to the requirement for maintaining multiple small-molecule inhibitors during ART to control viral replication [57]. The development of an autologous hematopoietic stem cell therapy, which could improve transplantation safety, enhance treatment efficacy by providing resistance to both dual- and CXCR4-tropic HIV, and expand the pool of HIV patients eligible for HSCT.
The CRISPR/Cas9 technology including base editing, prime editing, and zinc-finger nucleases—have emerged as promising approaches with distinct advantages in specificity, efficiency, and reduced off-target effects. These methods could provide solutions to challenges associated with conventional gene therapy, such as immune responses, delivery efficiency, and long-term genomic stability.
The diverse applications of this technology including in vivo, in vitro, and ex vivo approaches, highlighting its versatility in medical research and evaluating key limitations that may impact its future clinical translation.”
- Line 336: I have provided further insights into how targeting follicular dendritic cells could be improved, particularly through combination therapies or innovative CAR-based designs (page 12; line 566, new reference added [62], in purple).
Text added:
“Recent studies suggest that increasing dendritic cells recruitment and function could enhance T cell responses and improve therapeutic efficacy. Combination strategies that integrate dendritic cell vaccines, chimeric antigen receptor CAR T cells, and immune checkpoint inhibitors have demonstrated promise in preclinical and clinical studies. Notably, glioblastoma patients treated with dendritic cell vaccines in conjunction with Treg depletion, anti-PD-1 therapy, and adjuvants such as Poly I:C have shown extended progression-free survival [62]. These findings underscore the necessity of combination therapies to overcome glioblastoma’s immune barriers. Therefore, targeting follicular dendritic cells should be approached through strategies that not only enhance their antigen-presenting function but also promote a favorable immune microenvironment. This may involve optimizing CAR designs to engage both dendritic cells and T cells, utilizing adjuvants that support dendritic cell maturation and activation, and incorporating checkpoint blockade to counteract immunosuppression [62]. A multifaceted approach that restores the dendritic cell–T cell axis will likely be essential to achieving durable immune responses in glioblastoma treatment [62].”
I sincerely appreciate the reviewer's thorough assessment of my manuscript and their valuable suggestions to enhance its clarity, completeness, and scientific rigor. Below, I address each comment in detail:
- UNAIDS Fact Sheet 2024: I acknowledge the importance of incorporating the most recent epidemiological data. I have updated the manuscript with the latest statistics on HIV prevalence from the UNAIDS 2024 fact sheet (page 1; line 24, new reference added [1], in purple).
Text added:
“The UNAIDS 2024 Fact Sheet provides updated statistics on HIV prevalence of 2023, approximately 39.9 million individuals globally were living with HIV in that year, there were 1.3 million new HIV infections, and 630,000 people died from AIDS-related illnesses. Additionally, 30.7 million people were accessing antiretroviral therapy (ART) [1].”
- Clarification of ART and Latent Reservoirs: I appreciate this suggestion and have revised the relevant sections to merge overlapping statements, ensuring a more concise and coherent discussion, particularly before addressing significant breakthroughs in HIV research. (Page 3; line 96, new reference added [8], in purple).
Text added:
“LRAs, on the other hand, are drugs that reactivate viral transcription in the latently infected cells within the reservoir exposing these cells to immune clearance mechanisms or cytotoxic agents [8]. LRAs with demonstrated potency in cells from people living with HIV-1 have been hypothesised to reactivate latency through different mechanisms of action [8].”
- Definition of Latent Reservoirs: I have expanded the explanation of latent reservoirs to enhance clarity for a broader readership, providing a more detailed definition to improve comprehension (page 3; line 111, new references added [12-14]; line 141, new references added [15,16], in purple).
Text added:
“LRAs can be categorized into several classes based on their mechanisms of action. The first class includes epigenetic modifiers, which reverse the repressive epigenetic marks around the integrated provirus that influence HIV transcription. The histone methyltransferase (HMT) inhibitors and histone deacetylase inhibitors (HDACi) are the more studied inhibitors, which reverse the repressive acetyl and methyl marks on the integrated HIV genome, its surrounding DNA, and the associated histone tails in nucleosomes [12, 13]. BRG1/BRM-associated factor (BAF) inhibitors alter the positioning of nucleosomes on integrated HIV DNA, facilitating transcription of the HIV genome [14].”
“The second class, intracellular signaling modulators, consists of drugs that regulate protein kinases in signaling pathways, influencing the binding of transcription factors (TFs) to the long terminal repeat (LTR). These include protein kinase C (PKC) agonists and compounds within the PI3K/Akt or JAK/STAT pathways [13]. Additionally, second mitochondria-derived activator of caspases (SMAC) mimetics can be employed to inhibit the degradation of NF-κB-inducing kinase (NIK), thereby promoting the accumulation of NF-κB [15]. Other class of LRAs includes cytokine or immune receptor agonists, which stimulate immune cells through interleukins (ILs), cytokines, T cell receptors (TCRs), checkpoint inhibitors, or Toll-like receptor (TLR) agonists. Once transcription is initiated, transcription elongation factors can be used to enhance the activity of Tat, which is crucial for transcription elongation [16]. Notable examples in this class include BET inhibitors, which antagonize the inhibitor of P-TEFb, thereby activating the recruitment of P-TEFb to the LTR [12]. Finally, an unclassified group of LRAs includes previously used drugs that have been found to reactivate HIV, although their mechanisms of action remain unclear [13]. “
- CRISPR/Cas9 Niche Specification: I recognize the importance of specifying the niche that CRISPR/Cas9 technology addresses. The revised manuscript now explicitly discusses its contribution and targeted mechanisms in the context of HIV-1 cure strategies.
Text added:
CRISPR/Cas9 has been used to excise latent HIV Deoxyribonucleic acid (DNA) integrated into the host cell genome, representing a unique strategy to eradicate the virus rather than simply suppress it with ART.
Ongoing challenges in the scientific progress of the research field include addressing the evolution of viral resistance, immune evasion, and the heterogeneity of patient responses to treatments complicate the development of universally effective therapies. Collaborative research efforts are needed to overcome these hurdles and explore innovative therapeutic approaches, including gene editing technologies like CRISPR/Cas9, immune modulation, and combination therapies.
- Ongoing Challenges in Scientific Progress have been incorporated to highlight persisting challen that must be addressed for the effective implementation of emerging strategies (page 2; line 70; page 10; line 460, new references added [54-57]; page 12; line 599; page 16; line 792, new reference added [97]; page 18; line 861, new references added [101-107] in purple, blue, red and blue).
Text added:
“Ongoing challenges in the scientific progress of the research field include addressing the evolution of viral resistance, immune evasion, and the heterogeneity of patient responses to treatments complicate the development of universally effective therapies. Collaborative research efforts are needed to overcome these hurdles and explore innovative therapeutic approaches, including gene editing technologies like CRISPR/Cas9, immune modulation, and combination therapies.”
“While promising, this strategy is associated with significant risks and limitations. In the context of medical conditions, the primary challenges include disease relapse, infectious complications, and regimen-related toxicities, particularly among older patient populations [54]. Careful consideration must be given to quality of life and long-term outcomes, especially in elderly individuals. Access to HSCT is influenced by socioeconomic status, education level, and ethnicity, with migrants and minority groups frequently encountering systemic barriers [54]. Additionally, cultural beliefs and language barriers may hinder patient comprehension and adherence to HSCT protocols. Immigrant populations often exhibit distinct perceptions of illness and treatment, which can impact their interaction with and utilization of healthcare systems [54].
HSCT raises significant ethical concerns, particularly in the domains of informed consent, donor-recipient matching, and the long-term implications of transplantation. The decision-making process for HSCT is inherently complex, involving multiple stakeholders, including patients, families, and healthcare providers [55]. This process necessitates a careful balance between potential risks, clinical benefits, and the overall impact on the patient's quality of life. Pediatric HSCT introduces additional ethical challenges, particularly concerning the involvement of minors in decision-making processes and the long-term physical, psychological, and social effects on survivors. A particularly contentious issue is the right of mature minors to refuse life-sustaining treatments, such as HSCT, which often intersects with legal, ethical, and familial considerations. Furthermore, conflicts of interest in HSCT research and clinical practice may arise from various sources, including economic pressures, selective publication of data, and the prioritization of research that is financially lucrative over studies addressing unmet medical needs [55]. These conflicts can undermine the integrity of research and the equitable delivery of care. Ethical oversight committees play a pivotal role in addressing these challenges by safeguarding the rights and safety of research participants. These committees are responsible for ensuring compliance with internationally recognized ethical guidelines, such as the Declaration of Helsinki [55]. To maintain their credibility and effectiveness, ethical committees must operate independently and ensure that the benefits and risks of research are equitably distributed across all societal groups, thereby promoting justice and inclusivity in scientific advancement.
HSCT is a highly invasive and morbid procedure, often associated with graft-versus-host disease and challenges in finding HLA-compatible CCR5Δ32/Δ32 donors [46]. Furthermore, the strategy does not protect against HIV variants utilizing the CXCR4 co-receptor for cell entry, leaving the potential for continued viral replication [56]. These challenges underscore the need for careful consideration and further research to improve the safety and feasibility of this approach.
Patients infected with CXCR4-tropic HIV generally exhibit a poorer clinical prognosis and would not benefit from the transplantation of CCR5Δ32 hematopoietic stem and progenitor cells, as the CXCR4-tropic virus can independently infect cells without relying on the CCR5 co-receptor [57]. There is a critical need for strategies to combat the multiple variants of HIV that evolve in each patient, as well as the identification of therapies effective against both CCR5- and CXCR4-tropic HIV-1. Developing diverse genetic resistance mechanisms is comparable to the requirement for maintaining multiple small-molecule inhibitors during ART to control viral replication [57]. The development of an autologous hematopoietic stem cell therapy, which could improve transplantation safety, enhance treatment efficacy by providing resistance to both dual- and CXCR4-tropic HIV, and expand the pool of HIV patients eligible for HSCT.
The CRISPR/Cas9 technology including base editing, prime editing, and zinc-finger nucleases—have emerged as promising approaches with distinct advantages in specificity, efficiency, and reduced off-target effects. These methods could provide solutions to challenges associated with conventional gene therapy, such as immune responses, delivery efficiency, and long-term genomic stability.
The diverse applications of this technology including in vivo, in vitro, and ex vivo approaches, highlighting its versatility in medical research and evaluating key limitations that may impact its future clinical translation.”
“While substantial challenges remain, ongoing research and technological progress are steadily addressing these obstacles. Continued exploration of novel strategies, combined with rigorous clinical evaluation, will be essential for fully realizing the therapeutic potential of these approaches in combating HIV-1.”
“Despite promising in vitro results, translating these findings into in vivo models has presented challenges. In a study involving simian-human immunodeficiency virus (SHIV)-infected monkeys, the administration of dual-affinity re-targeting molecules targeting HIV-1 Env and CD3, in combination with a latency-reversing agent, did not lead to a significant reduction in latently infected cells. Moreover, the treatment induced the production of anti-drug antibodies, resulting in a rapid decline in serum concentration after multiple infusions [97]. Overall, bispecific antibodies offer a promising avenue for HIV-1 cure strategies by facilitating the targeted elimination of infected cells through immune activation. However, further research is necessary to address the challenges observed in in vivo applications and to optimize their therapeutic potential.”
“Advancements in single-cell technologies have significantly transformed our comprehension of the mechanisms underlying HIV-1 persistence, as well as contributing to the fields of human diseases, immunology, oncology, and development. A key contribution to the HIV-1 field is the application of single-cell multi-omics, which is one of the few methods capable of addressing the heterogeneity and rarity of HIV-1-infected cells, especially those lacking specific markers for enrichment [101]. Genome-wide profiling offers an unbiased means to identify various cell types and uncover novel mechanisms. The integration of multi-omic profiling, encompassing epigenetic regulators (DNA via Assay for Transposase-Accessible Chromatin with high-throughput sequencing), transcriptional networks (RNA via RNA-sequencing), and cellular markers/therapeutic targets (proteins via Cellular Indexing of Transcriptomes and Epitopes by sequencing), enhances our understanding of the HIV-1 reservoir through the lens of the central dogma of molecular biology. TCR clonal tracking reveals the clonal dynamics of HIV-1-infected cells. It is essential to exercise caution to ensure that findings are mechanistically meaningful, biologically reproducible, and statistically sound [101]. Bioinformatic analysis of single-cell multi-omics must be tailored to the specific biological questions at hand, rather than relying on default protocols. Accurate identification of rare HIV-1+ cells demands meticulous procedures to prevent sequencing and mapping artifacts [101].
In addition to the use in predictive modeling through machine learning, artificial intelligence has been employed to enhance HIV serodisclosure efforts. In a small pilot study conducted by Muessig et al., developed and assessed the Tough Talks virtual reality program designed to assist young men who have sex with men in role-playing HIV serostatus disclosure, aiming to promote protective behaviors against HIV transmission [102]. The study gathered qualitative data through focus groups with young men who have sex with men living with HIV, exploring their experiences with serodisclosure, which were then used to create a database of common expressions encountered during HIV serostatus discussions. Participants were able to select a virtual character (i.e., an avatar) and engage in various disclosure scenarios using the virtual reality platform. Most participants found the tool acceptable, although some considered it overly complex and cumbersome. A randomized trial is planned to assess the impact of this artificial intelligence tool, comparing its effects when delivered online versus in a clinic setting, on HIV viral load and condomless anal sex behaviors among young men who have sex with men [103]. Chatbots, also known as conversational agents, have been a relatively underexplored application of artificial intelligence in HIV prevention thus far. These chatbots can interact with users anonymously through voice or text messaging, utilizing machine learning algorithms to generate context-appropriate prompts or responses based on prior interactions. The rapid advances in natural language processing, mobile applications and social media platforms have significantly accelerated the adoption of chatbots [104]. Brixey et al., in 2016, launched a chatbot for sexual health information related to HIV/AIDS (SHIHbot) via Facebook Messenger, offering users information sourced from a database compiled from professional medical and public health materials [105]. The U.S. Department of Health and Human Services piloted, in 2018, a chatbot on Facebook Messenger during the International AIDS Conference, gaining valuable insights into the need for tailored, conversational, and multimedia content for the future development of chatbots aimed at HIV prevention [106]. Beyond delivering HIV-related information, chatbots hold potential for supporting individuals in decisions regarding preexposure prophylaxis use or adherence to preexposure prophylaxis and antiretroviral therapy, though these applications remain unexplored [107].
Ultimately, continual learning from disciplines beyond HIV-1 research, including artificial intelligence, bioinformatics, biotechnology, the human cell atlas, immunology, and cancer biology, will expedite groundbreaking discoveries regarding the mechanisms of HIV-1 persistence and the development of novel therapeutic strategies.
- Consistent Formatting of Terminology: I have carefully reviewed the manuscript to ensure consistency in terminology, and particularly in explaining terms, such as "BRG1/BRM-associated factor." (Page 3; line 117, new reference added [14], in purple and blue).
Text added:
“BRG1/BRM-associated factor (BAF) inhibitors alter the positioning of nucleosomes on integrated HIV DNA, facilitating transcription of the HIV genome [14]. In an interesanting study shows these BAF complexes facilitate the access of transcription factors and cofactors to enhancers and promoters, thereby regulating gene expression critical for the growth and differentiation of acute myeloid leukemia stem and progenitor cells. In acute myeloid leukemia with MLL1 rearrangement (MLL1r) or mutant NPM1 (mtNPM1), treatment with menin inhibitors (MIs) can induce clinical remissions; however, many patients either exhibit resistance or experience relapse, with some acquiring menin mutations. FHD-286, an orally bioavailable and selective BRG1/BRM inhibitor currently in clinical development for acute myeloid leukemia, has demonstrated potent effects in preclinical studies. FHD-286 induces differentiation and lethality in AML cells with MLL1r or mtNPM1 by disrupting chromatin accessibility and downregulating key factors such as c-Myc, PU.1, and CDK4/6. “
- Expansion on the Block and Lock Strategy: I have elaborated on the molecular mechanisms underlying the "block and lock" strategy, detailing transcriptional silencing pathways.
Text added:
“ It can indeed be further elaborated by delving into the molecular mechanisms of transcriptional silencing and this topic, which will be addressed later, will include examples from scientific studies.. This strategy involves the use of specific molecules or genetic modifications to lock the virus in a transcriptionally silent state within the host cell. At the molecular level, it typically targets key transcription factors or epigenetic regulators that maintain HIV latency. By enhancing the understanding of how these molecular mechanisms, such as chromatin remodeling, DNA methylation, and histone modifications, contribute to silencing viral gene expression, researchers can identify more precise targets to sustain the latent state and prevent viral reactivation. This could lead to improved therapeutic strategies for HIV eradication.”
- Limitations and Therapeutic Considerations: I have expanded the discussion of LEDGINs in the section of “block and lock” approach key limitations, including the therapeutic window, resistance development, and potential approaches to mitigate these challenges (page 7; line 302, new reference added [37], in purple).
“For LEDGINs, their efficacy is contingent on early administration, as their therapeutic window is limited [37]. A study employed a single-cell branched DNA imaging technique to simultaneously detect viral DNA and RNA, allowing for a detailed assessment of the impact of LEDGIN treatment on HIV-1 integration, transcription, and reactivation in both cell lines and primary cells. These findings demonstrated that LEDGIN-mediated retargeting reduces basal transcriptional activity and impairs proviral reactivation, as evidenced by a significant decrease in viral RNA expression per residual provirus. The interaction between HIV-1 integrase and the chromatin tethering factor LEDGF/p75 serves as a key determinant of integration site preference [37]. By employing LEDGINs to disrupt this interaction, researchers have elucidated how integration retargeting influences the three-dimensional positioning of the provirus, its transcriptional activity, and its susceptibility to reactivation [37]. These results support the feasibility of a "block-and-lock" strategy aimed at permanently silencing HIV-1 by directing integration into genomic regions refractory to reactivation following ART discontinuation.”
- Addressing Data Gaps and Future Research Directions: I sincerely appreciate your valuable suggestion regarding the inclusion of outcomes that future studies should aim to address, such as long-term efficacy, resistance mechanisms, and biomarkers for monitoring viral reservoirs. These aspects have already been addressed in the manuscript as part of the discussion of each therapeutic strategy. Additionally, I have been included a new dedicated section outlining an additional strategy (7. Bispecific antibodies method)
- Strengthening the Conclusion: To reinforce the conclusion, I have changed this section and explicitly highlighted the balance between existing ART regimens and emerging cure strategies, emphasizing their potential to pave the way for a functional cure.
I greatly appreciate the reviewer's insightful comments, which have significantly improved the manuscript. Thank you for your time and thoughtful suggestions.

Round 2
Reviewer 3 Report
Comments and Suggestions for Authors
The authors have responded to all the reviewer inquiry and the manuscript has been significantly improved and now may be publishable